# TRACE ANYTHING: REPRESENTING ANY VIDEO IN 4D VIA TRAJECTORY FIELDS

**Xinhang Liu**
ByteDance Seed
HKUST

**Yuxi Xiao**
ByteDance Seed
Zhejiang University

**Donny Y. Chen**
ByteDance Seed

**Jiashi Feng**
ByteDance Seed

**Yu-Wing Tai**
Dartmouth College

**Chi-Keung Tang**
HKUST

**Bingyi Kang**
ByteDance Seed

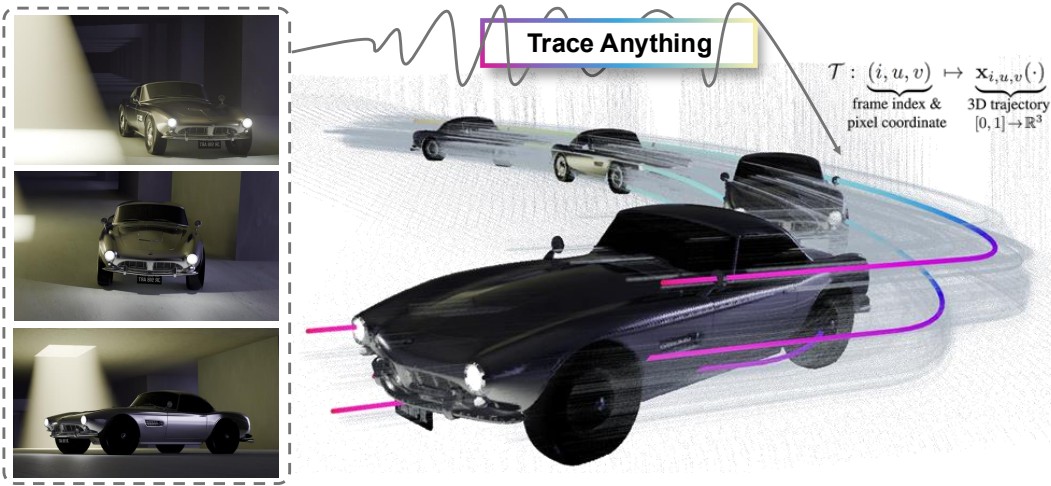

**Input Frames**          **Trajectory Field**

Figure 1: Any video* can be represented in 4D with a **Trajectory Field**, a dense mapping assigning each pixel in each frame to a parametric 3D trajectory. We propose **Trace Anything**, a neural network that predicts the trajectory field with a single forward pass.

## ABSTRACT

Building 4D video representations to model underlying spacetime constitutes a crucial step toward understanding dynamic scenes, yet there is no consensus on the paradigm: current approaches resort to additional estimators such as depth, flow, or tracking, or to heavy per-scene optimization, making them brittle and hard to generalize. In a video, its atomic unit, the pixel, follows a continuous 3D trajectory that unfolds over time, acting as the atomic primitive of dynamics. Recognizing this, we propose to represent any video* as a *Trajectory Field*: a dense mapping that assigns each pixel in each frame to a parametric 3D trajectory. To this end, we introduce *Trace Anything*, a neural network that predicts the trajectory field in a feed-forward manner. Specifically, for each video frame, the model outputs a series of control point maps, defining parametric trajectories for each pixel. Together, our representation and model directly construct a 4D video representation in a single forward pass, without additional estimators or global alignment. We develop a *synthetic data platform* to construct a training dataset and a benchmark for trajectory field estimation. Experiments show that Trace Anything surpasses existing methods or performs competitively on the new benchmark and established point tracking benchmarks, with significant efficiency gains. Moreover, it facilitates downstream applications such as goal-conditioned manipulation, simple motion extrapolation, and spatio-temporal fusion. We will release the code and the model weights.

---

*Here, "any video" extends beyond monocular videos to include image pairs or even unordered unstructured image collections that capture dynamic scenes.

## 1 INTRODUCTION

Understanding dynamic scenes requires more than disjoint reconstruction of 3D space at each time step; it demands modeling how the scene evolves in both space and time. A central challenge toward spatial intelligence Yang et al. (2025c;b); Xiao et al. (2025a) is to develop a 4D video representation that captures the underlying spacetime dynamics in a way that is geometrically grounded and scalable. Rather than relying on additional estimators such as depth, flow, or tracking, or on heavy per-scene optimization, we observe that the atomic elements of video, its pixels, naturally trace out *3D trajectories* in the world, which acts as the atomic primitive of dynamics.

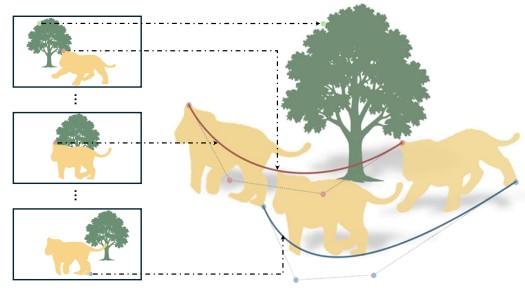

Input Frames   Per-Pixel 3D Parametric Trajectories

Figure 2: Given the input frames (left), a *trajectory field* represents the video at the atomic level, mapping each pixel in each frame to a 3D trajectory, expressed as a parametric curve (right).

Recognizing this, we propose **Trajectory Fields**, a versatile 4D representation for any video that associates each pixel in each frame with a parametric 3D trajectory, as illustrated in Figure 2. Unlike prior 4D reconstruction methods that produce disjoint per-frame point clouds (Zhang et al., 2025a; Li et al., 2025; Sucar et al., 2025) and rely on estimated optical flow or 2D tracks to build cross-frame correspondences, Trajectory Fields offer a more direct and compact way to model scene dynamics.

Building on this representation, we propose **Trace Anything**, a feed-forward neural network that estimates trajectory fields directly from video frames. As shown in Figure 1, with a single forward pass over all input frames, it predicts a stack of control point maps for each frame, defining spline-based parametric trajectories for every pixel. This design brings three key advantages: (i) its one-pass scheme eliminates intermediate estimators and iterative global alignment, (ii) it predicts all trajectories (per pixel per frame) jointly in a shared world coordinate system, and (iii) it generalizes across diverse inputs, including monocular videos, image pairs, and unordered photo sets.

To support training and evaluation at scale, we develop a Blender-based platform featuring diverse environments, moving characters, and camera trajectories. It produces photo-realistic renderings with dense annotations, including 2D/3D trajectories, depth, semantics, flow, and camera poses. From this platform, we release (i) the *Trace Anything dataset*—10,000+ videos (120 frames each) for training trajectory field estimation models, and (ii) the *Trace Anything benchmark*—200 curated videos for evaluating models' ability to capture motion jointly across all frames.

Trained with our new dataset, Trace Anything achieves state-of-the-art results on our trajectory field benchmark and performs competitively on established point tracking benchmarks, while offering significant efficiency gains. Moreover, our paradigm facilitates downstream spatial reasoning applications, including simple motion extrapolation, spatio-temporal fusion, and goal-conditioned manipulation.

In summary, our contributions are:

- We propose **Trajectory Fields** as an atomic-level and versatile 4D video representation, grounded in a principled formulation.
- We present **Trace Anything**, a feed-forward network that predicts trajectory fields without requiring extra estimators or per-scene optimization.
- We develop a synthetic data platform for large-scale training and benchmarking of trajectory field estimation.
- Extensive experiments on existing and new benchmarks demonstrate competitive accuracy, faster inference, and new capabilities.

## 2 RELATED WORK

**(Dynamic) 3D scene reconstruction.** Reconstructing 3D structure from multi-view images is a long-standing problem in computer vision. Classical Structure-from-Motion (SfM) pipelines (Hart-

ley & Zisserman, 2003; Agarwal et al., 2011; Schonberger & Frahm, 2016) proceed in sequential stages: feature extraction, image matching, triangulation, relative pose estimation, and global bundle adjustment. Deep learning has improved individual components (DeTone et al., 2018; Sarlin et al., 2020) yet stage-wise pipelines remain prone to error accumulation. DUSt3R (Wang et al., 2024a) addressed this by directly predicting 3D pointmaps from image pairs. Fast3R (Yang et al., 2025a), VGGT (Wang et al., 2025a), $\pi^3$ (Wang et al., 2025b) and MapAnything (Keetha et al., 2025) further relaxed the pairwise assumption with all-to-all attention, enabling joint reasoning over all frames and avoiding $O(N^2)$ pairwise inference. However, both traditional and learning-based methods generally assume static scenes and sufficient camera baselines, leading to degraded performance in dynamic settings. To handle monocular videos with dynamics, MegaSAM (Li et al., 2025) integrates optimization-based SLAM, while Monst3R (Zhang et al., 2025a), POMATO (Zhang et al., 2025b), Easi3R Chen et al. (2025), St4RTrack (Feng et al., 2025), and Dynamic Point Maps (Sucar et al., 2025) extend DUSt3R-style networks to dynamic scenes. These methods typically generate disjoint per-frame point clouds, relying on optical flow or 2D tracks for cross-frame correspondences, and their pairwise inference often requires costly per-scene optimization for global alignment. In contrast, Trace Anything directly estimates trajectory fields that produce dynamic point clouds with cross-frame correspondences, sharing the feed-forward spirit of Yang et al. (2025a) and Wang et al. (2025a) and performing one-pass inference over all frames.

**Point tracking.** Particle Video (Sand & Teller, 2008) first introduced long-range particle trajectories in videos. Early deep learning methods (Harley et al., 2022; Doersch et al., 2022; 2023) approached this with global matching and local refinement. CoTracker (Karaev et al., 2024b) leveraged a transformer-based architecture to enable tracking through occlusions, followed by works (Li et al., 2024a; Cho et al., 2024) that improved efficiency with 4D correlation volumes. CoTracker3 (Karaev et al., 2024a) further leveraged unlabeled data to boost performance. 3D point tracking remains comparatively new. OmniMotion (Wang et al., 2023) addressed the task via test-time optimization, while SpatialTracker (Xiao et al., 2024) introduced the first feed-forward 3D tracker by combining 2D tracking with monocular depth priors. DELTA (Ngo et al., 2025) achieved dense 3D tracking using a transformer with upsampling for high-resolution outputs. Concurrently, SpatialTrackerV2 (Xiao et al., 2025b) scaled training across real and synthetic data, and St4RTrack (Feng et al., 2025) and POMATO Zhang et al. (2025b) extended 3D reconstruction models for tracking via joint optimization. Unlike prior approaches, our method bypasses monocular depth estimation and 2D trackers and directly predicts dense 3D trajectories in a feed-forward manner.

**4D representations for NVS.** A large class of 4D representations has been developed for novel view synthesis (NVS) in dynamic scenes, aiming to deliver immersive effects such as "bullet time." Since Neural Radiance Fields (NeRF) (Mildenhall et al., 2020) introduced implicit volumetric representations, many extensions have incorporated temporal dynamics. One class of methods (Gao et al., 2021; Li et al., 2021a;b; Xian et al., 2021) directly conditions the radiance field on time, treating density and color as continuous functions of space and time. Another class (Pumarola et al., 2021; Zhang et al., 2021; Park et al., 2021a;b; Tretschk et al., 2021) maps observations to a canonical space and models dynamics via deformation fields. Grid-based approaches (Cao & Johnson, 2023; Fridovich-Keil et al., 2023; Wang et al., 2022; Attal et al., 2023; Liu et al., 2024) discretize the 4D volume into compact planar factors for efficiency. Also in this line of work, Wang et al. (2021) proposed 'neural trajectory fields', with a different formulation and purpose than 'trajectory fields' in this study. More recently, 3D Gaussian Splatting (3DGS) (Kerbl et al., 2023) has been extended to dynamics (Wu et al., 2024; Yang et al., 2023; Luiten et al., 2023; Yang et al., 2024; Li et al., 2024b), improving rendering quality and speed. These efforts focus on photorealistic appearance and typically assume precomputed camera poses or point clouds. Our work is orthogonal: we propose a geometry-centric paradigm that directly infers trajectory fields from raw videos, emphasizing accurate 3D motion modeling. Integrating NVS with our paradigm, e.g., using trajectory fields to initialize dynamic 3DGS models, is a promising future direction.

## 3 METHOD

The atomic elements of video, its pixels, naturally trace out 3D trajectories in the world, forming the primitive units of dynamics. Recognizing this, we aim to model dynamic scenes through trajectory fields, a 4D representation that encodes each pixel in each frame as a continuous 3D trajectory over time. In the following, we first formalize trajectory fields in Section 3.1, then present Trace Anything

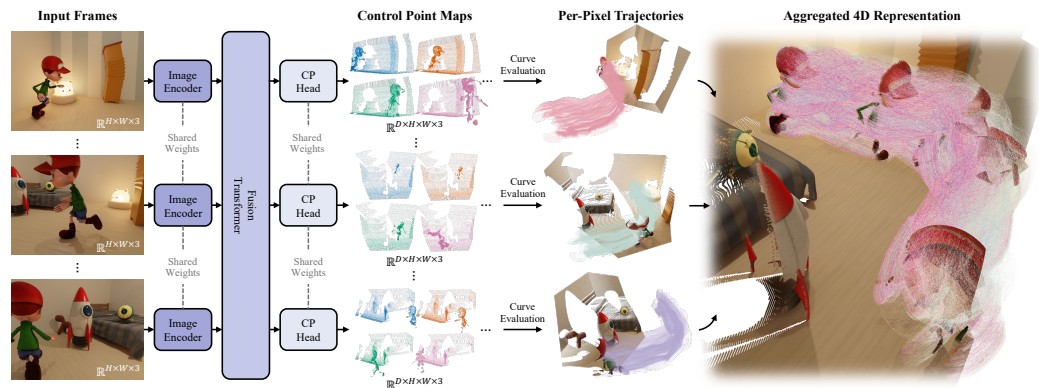

Figure 3: **Trace Anything pipeline.** Input frames are processed by a geometric backbone consisting of an image encoder and a fusion transformer. The control point head outputs dense control point maps $\mathbf{P}_i \in \mathbb{R}^{D \times H \times W \times 3}$, where $\mathbf{P}_{i,u,v}^{(k)}$ is the $k$-th control point for pixel $(u, v)$ in frame $I_i$. These define continuous 3D trajectories $\mathbf{x}_{i,u,v}(t)$ via cubic B-splines, yielding a 4D reconstruction.

in Section 3.2, a feed-forward network designed to estimate them, and finally describe the overall training scheme in Section 3.3. In this section, we define a *field* as a mapping from a domain $M$ to a codomain $V$, $\mathcal{F} : M \to V$, where $M$ may be a discrete or continuous space, and $V$ may represent scalars, vectors, or functions. We provide preliminaries on fields in Section A and on parametric curves in Section B.

## 3.1 PROBLEM FORMULATION

We formalize *trajectory fields*, a 4D representation of dynamic 3D scenes in a video. Let $\{I_i\}_{i=1}^N$ be a collection of $N$ RGB frames, where each $I_i \in \mathbb{R}^{3 \times H \times W}$ captures the scene at different time steps or viewpoints. A trajectory field is defined as

$$\mathcal{T} : [N] \times [H] \times [W] \to C([0,1], \mathbb{R}^3), \quad (i, u, v) \mapsto \mathbf{x}_{i,u,v}(\cdot) \tag{1}$$

where $[N]$, $[H]$, and $[W]$ denote the discrete sets of frame indices and pixel coordinates, respectively, and $\mathbf{x}_{i,u,v} : [0,1] \to \mathbb{R}^3$ is a continuous 3D trajectory for pixel $(u, v)$ in frame $I_i$. The domain is $M = [N] \times [H] \times [W]$—unlike classical physical fields, our trajectory field is defined on a discrete pixel grid and imposes no spatial continuity The codomain is $V = C([0,1], \mathbb{R}^3)$, the space of continuous functions from $[0,1]$ to $\mathbb{R}^3$. Here, $t \in [0,1]$ is a normalized continuous time parameter spanning the full temporal extent. For video input, $t = 0$ corresponds to the first frame and $t = 1$ to the last frame.

Each trajectory $\mathbf{x}_{i,u,v}(t)$ is parameterized as a spline-based curve with $D$ control points, defined as

$$\mathbf{P}_i \in \mathbb{R}^{D \times H \times W \times 3}, \tag{2}$$

where $\mathbf{P}_{i,u,v}^{(k)} \in \mathbb{R}^3$ is the $k$-th control point for pixel $(u, v)$ in frame $I_i$, with $k \in \{0, 1, \ldots, D-1\}$. Each of the $N$ input frames contributes its own stack of $D$ control-point maps, so the video produces $N \times D$ control-point maps in total.

Given basis functions $\{\phi_k(t)\}_{k=0}^{D-1}$, the trajectory is

$$\mathbf{x}_{i,u,v}(t) = \sum_{k=0}^{D-1} \mathbf{P}_{i,u,v}^{(k)} \phi_k(t). \tag{3}$$

The form of the basis functions $\{\phi_k(t)\}_{k=0}^{D-1}$ depends on the type of parametric curve. In our implementation, we use cubic B-splines with clamped knots as detailed in Section B.

Figure 2 illustrates this formulation of trajectory fields. For any pixel from any frame, its 3D coordinate at any time $t \in [0,1]$ can be obtained with Equation (3). This fundamentally differs from existing 4D reconstruction methods that predict per-frame disjoint point clouds and establish cross-frame correspondences via estimated optical flow or 2D tracks. Ideally, two conditions should hold: (C1) pixels in static regions collapse to degenerate trajectories; (C2) corresponding pixels from different frames map to the same 3D trajectory.

## 3.2 Network Architecture

Building on the formulation in Section 3.1, we propose a feed-forward network, *Trace Anything* (Figure 3), which predicts trajectory fields directly from video or unstructured image sets. For each frame, it outputs control point maps defining parametric curves over time, enabling trajectory field estimation in a single pass. This design eliminates reliance on depth or optical flow and avoids per-scene iterative optimization, providing a compact, efficient approach to modeling 4D scenes.

**Geometric backbone.** We build Trace Anything upon a feed-forward geometric backbone, similar in spirit to recent models (Wang et al., 2025a; Yang et al., 2025a). Each frame is first tokenized by an image encoder, followed by a fusion transformer that integrates spatio-temporal context across views through interleaved frame-wise and global attention layers. For sequential video input, we additionally inject temporal index embeddings, while the architecture remains compatible with unordered or unstructured image collections.

**Control Point Head.** Built on the backbone features, the *control point head* outputs dense control point maps $\mathbf{P}_i \in \mathbb{R}^{D \times H \times W \times 3}$ for each input frame $I_i$. Each pixel $(u,v)$ has $D$ control points $\{\mathbf{P}_{i,u,v}^{(k)}\}_{k=0}^{D-1}$, compactly parameterizing its 3D trajectory. Predictions are in a shared world coordinate system, with an optional *local CP head* estimating control points in each frame's local camera system. The head also predicts *per-control-point confidence scores* $\Sigma_{i,u,v}^{(k)}$ for confidence-weighted training and filtering uncertain estimates at inference.

**Curve evaluation.** Given the predicted control points and basis functions $\{\phi_k(t)\}_{k=0}^{D-1}$, continuous trajectories $\mathbf{x}_{i,u,v}(t)$ are obtained via Equation (3). At evaluation time, the trajectory can be queried at any $t \in [0,1]$. In particular,

$$\mathbf{x}_{i,u,v}(0) = \sum_{k=0}^{D-1} \mathbf{P}_{i,u,v}^{(k)} \cdot \phi_k(0) \overset{*}{=} \mathbf{P}_{i,u,v}^{(0)}, \quad \mathbf{x}_{i,u,v}(1) = \sum_{k=0}^{D-1} \mathbf{P}_{i,u,v}^{(k)} \cdot \phi_k(1) \overset{*}{=} \mathbf{P}_{i,u,v}^{(D-1)}, \quad (4)$$

where $(*)$ holds for cubic B-splines with clamped knots or for Bézier bases.

To obtain the 3D coordinates of a pixel from frame $i$ evaluated at the acquisition time of another frame $j$, we substitute the corresponding temporal parameter $t_j$ into its trajectory:

$$\mathbf{X}_{i \rightarrow j}(u,v) = \mathbf{x}_{i,u,v}(t_j). \quad (5)$$

In most cases, $t_j$ is obtained from metadata or frame order. Otherwise, an auxiliary *timestamp head* predicts normalized timestamps $\hat{t}_j \in [0,1]$. As a special case, evaluating each trajectory at frame $i$'s own acquisition time $t_i$ recovers the 3D point map for frame $I_i$:

$$\mathbf{X}_i(u,v) = \mathbf{x}_{i,u,v}(t_i). \quad (6)$$

Trace Anything outputs the trajectory field with a single network inference for all frames, avoiding pairwise inference and subsequent global alignment, while being self-contained and independent of external estimators for monocular depth, optical flow, or 2D tracks.

## 3.3 Training Scheme

To train Trace Anything, we directly supervise the accuracy of predicted trajectories. Intuitively, a trajectory predicted from frame $i$ should, when evaluated at the timestamp of another frame $j$, land exactly at its ground-truth 3D location at frame $j$'s acquisition time.

**Trajectory loss.** For a pixel $(u,v)$ in frame $i$, the predicted 3D position evaluated at $t_j$ is $\mathbf{X}_{i \rightarrow j}(u,v)$ (Equation (5)), while the corresponding ground truth is $\mathbf{X}_{i \rightarrow j}^{\text{gt}}(u,v)$. We define the loss as

$$\ell_{i \rightarrow j}(u,v) = \left\| \mathbf{X}_{i \rightarrow j}(u,v) - \mathbf{X}_{i \rightarrow j}^{\text{gt}}(u,v) \right\|_2^2. \quad (7)$$

**Confidence adjustment.** To account for the varying reliability of predicted trajectories across pixels and control points, we incorporate confidence adjustment. For each control point, the network predicts a scalar confidence $\hat{\Sigma}_{i,u,v}^{(k)} > 0$ alongside its 3D coordinates. The confidence associated with $\mathbf{X}_{i \rightarrow j}(u,v)$ is then aggregated using the same basis functions as in Equation (3):

$$\hat{\Sigma}_{i \rightarrow j}(u,v) = \sum_{k=0}^{D-1} \hat{\Sigma}_{i,u,v}^{(k)} \cdot \phi_k(t_j). \quad (8)$$

The final confidence-adjusted loss then becomes

$$\mathcal{L}_{\text{traj-conf}} = \frac{1}{|\Omega|} \sum_{(i,j)} \sum_{(u,v) \in \Omega} \left[ \hat{\Sigma}_{i \to j}(u,v)\, \ell_{i \to j}(u,v) + \alpha \log \hat{\Sigma}_{i \to j}(u,v) \right], \tag{9}$$

where $\Omega$ denotes valid pixels with ground-truth supervision. This adjustment downweights uncertain predictions while discouraging overconfident ones.

**Timestamp supervision.** When ground-truth timestamps are available, we directly supervise *Timestamp Head* with an $L_1$ regression loss:

$$\mathcal{L}_{\text{time}} = \frac{1}{N} \sum_{i=1}^{N} |\hat{t}_i - t_i|. \tag{10}$$

**Static regularization.** To encourage condition (C1), pixels in static regions should map to overlapped 3D control points. We enforce this by minimizing the variance of their control points:

$$\mathcal{L}_{\text{static}} = \frac{1}{|\Omega_{\text{static}}|} \sum_{(i,u,v) \in \Omega_{\text{static}}} \text{Var}\left( \{\mathbf{P}_{i,u,v}^{(k)}\}_{k=0}^{D-1} \right). \tag{11}$$

**Rigidity regularization.** For pixels segmented as belonging to the same rigid region, their trajectories should preserve internal distances across control points. Equivalently, the pairwise distance between any two pixels $p, q$ within a rigid segment should remain constant across control points. We enforce this by minimizing the variance of their distances:

$$\mathcal{L}_{\text{rigid}} = \frac{1}{|\Omega_{\text{rigid}}|} \sum_{(p,q) \in \Omega_{\text{rigid}}} \text{Var}\left( \{\|\mathbf{P}_p^{(k)} - \mathbf{P}_q^{(k)}\|_2\}_{k=0}^{D-1} \right). \tag{12}$$

**Correspondence regularization.** To encourage condition (C2), pixels with known cross-frame correspondences should share identical control points. Let $\Omega_{\text{corr}}$ be the set of matched pixels $((i,u,v), (j,u',v'))$. We penalize discrepancies between their control-point sequences:

$$\mathcal{L}_{\text{corr}} = \frac{1}{|\Omega_{\text{corr}}|} \sum_{\substack{((i,u,v), \\ (j,u',v')) \in \Omega_{\text{corr}}}} \frac{1}{D} \sum_{k=0}^{D-1} \left\| \mathbf{P}_{i,u,v}^{(k)} - \mathbf{P}_{j,u',v'}^{(k)} \right\|_2^2. \tag{13}$$

**Final objective.** The overall loss combines the core trajectory supervision with the above regularization terms:

$$\mathcal{L} = \mathcal{L}_{\text{traj-conf}} + \lambda_{\text{time}} \mathcal{L}_{\text{time}} + \lambda_{\text{static}} \mathcal{L}_{\text{static}} + \lambda_{\text{rigid}} \mathcal{L}_{\text{rigid}} + \lambda_{\text{corr}} \mathcal{L}_{\text{corr}}. \tag{14}$$

## 4 TRACE ANYTHING DATA PLATFORM

Data-driven modeling of dynamic scenes is limited by the lack of large-scale datasets with dense, reliable annotations. Existing synthetic datasets and generators (Dosovitskiy et al., 2015; Mayer et al., 2016; Harley et al., 2022; Greff et al., 2022; Zheng et al., 2023) are typically small and biased toward rigid motion, with sparse or short-term annotations, which are insufficient for realistic scene understanding and diverse dynamics. To address this, we develop a scalable 4D Scene Data Platform in Blender that synthesizes photo-realistic dynamic scenes with dense ground-truth annotations.

**Trace Anything dataset.** Using our platform, we build a dataset whose primary purpose is to train the Trace Anything model on trajectory field estimation. The current release contains about 10K unique scenes, each with 120 annotated frames. The collection spans a wide range of settings and motions, with examples shown in Figure B. The dataset exhibits diversity across multiple aspects: *(i) Environment* — diverse indoor and outdoor backgrounds from public asset libraries and procedural generation (Raistrick et al., 2023; 2024); *(ii) Dynamics* — articulated human characters and movable objects with both rigid and non-rigid motion; *(iii) Camera motion* — smooth trajectories sampled around active regions to mimic natural filming. Rendered RGB videos are paired with per-pixel 2D/3D trajectories, depth maps, camera poses, semantic masks, which facilitate the training scheme introduced in Section 3.3. Since our platform is fully programmable, it can be easily extended with new assets, domains, or annotation modalities to support future research.

**Trace Anything benchmark.** To evaluate the task of trajectory field estimation, we construct a benchmark consisting of 200 videos, each with 120 frames. A key difference from established point

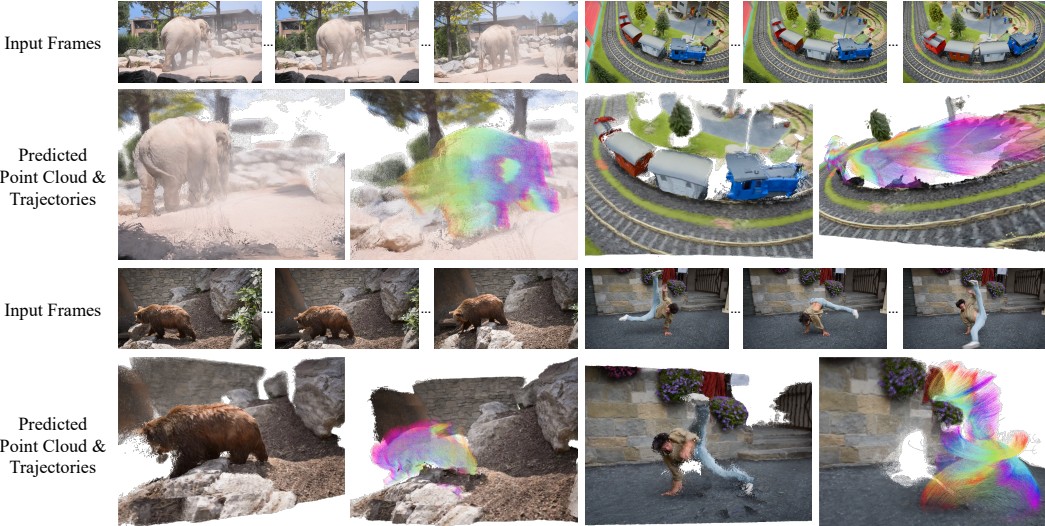

Figure 4: **Video-based trajectory field estimation on DAVIS (Perazzi et al., 2016).** Trace Anything predicts trajectory fields that can yield dynamic point cloud sequences and dense 3D trajectories, while remaining robust to complex non-rigid motion and occlusions.

tracking datasets (Koppula et al., 2024) lies in the evaluation protocol: point tracking benchmarks evaluate estimated trajectories only for pixels sampled from the first frame (*first-to-all*), whereas our benchmark evaluates trajectories for pixels sampled from *all* frames (*all-to-all*). This requires models not only to follow motion from a single starting frame, but also to jointly capture dynamics across the entire sequence. In addition, our benchmark provides denser trajectory annotations, covering more pixels per framel, and evaluates in world coordinates, requiring models to reason about both global geometry and motion.

# 5 EXPERIMENTS

In this section, we evaluate our method across a series of challenging settings, demonstrating its competitive accuracy, faster inference, and novel capabilities. Please refer to the appendix for implementation details and additional experimental results, and to the supplementary materials for videos and interactive demos.

## 5.1 TRAJECTORY FIELD ESTIMATION

We present qualitative results of trajectory field estimation on videos and image pairs. Further results on unstructured, unordered image sets are included in Section E, and qualitative comparisons are provided in Section E.4.

**Video to trajectory field.** For computational efficiency, we uniformly subsample long sequences to fewer than 60 frames. Figure 4 shows qualitative results on DAVIS (Perazzi et al., 2016), covering diverse dynamic scenes. Our predictions faithfully reconstruct both dynamic and static components of the scene, yielding dense, pixel-level 3D trajectories. These trajectories capture motions ranging from near-rigid transformations, such as a toy train moving along a track, to highly non-rigid deformations, such as humans or animals in motion, while also handling severe occlusions and preserving global scene structure.

**Image pair to trajectory field.** Our approach can also infer trajectory fields directly from image pairs, effectively reconstructing the implied spatio-temporal dynamics and interpolating intermediate motion. For this experiment, we use BridgeData V2 (Walke et al., 2023), a large and diverse dataset of robotic manipulation behaviors. Image pairs are sampled from video sequences with a temporal gap of 10–20 frames. As illustrated in Figure 5, given an initial image of a scene and a goal image specifying the desired outcome, our model predicts a trajectory field that captures plausible 3D trajectories of both objects and agents involved. These trajectories can also be re-projected with estimated camera poses to yield 2D trajectories (see Section E.3 and Figure J for details). In the context of robot learning, this naturally aligns with *goal-conditioned manipulation*, where predicted trajectories can be interpreted as feasible robot end-effector motions (Bharadhwaj et al., 2024).

Table 1: **Quantitative results on video-based trajectory field estimation.** CA is reported in $10^{-2}$ and SDD in $10^{-3}$. Best in **bold**, second-best underlined.

| Method | $\text{EPE}_{\text{mix}}\downarrow$ | $\text{EPE}_{\text{sta}}\downarrow$ | $\text{EPE}_{\text{dyn}}\downarrow$ | CA $\downarrow$ | SDD $\downarrow$ | $\text{APD}_{3D}\uparrow$ | AJ $\uparrow$ | Runtime (s) $\downarrow$ |
|---|---|---|---|---|---|---|---|---|
| CoTracker3 + VGGT | 0.518 | 0.461 | 0.555 | 7.83 | 1.67 | 18.8 | 13.8 | 197.4 |
| DELTA | 0.404 | 0.384 | 0.425 | 6.24 | 1.75 | 24.7 | 16.1 | 231.6 |
| SpaTrackerV2 | 0.296 | 0.291 | 0.366 | 7.24 | 1.51 | 25.5 | 16.8 | 178.4 |
| SpaTrackerV2* | 0.288 | 0.282 | 0.323 | 7.04 | 1.59 | 31.3 | **21.3** | 174.7 |
| MonST3R | 0.316 | 0.258 | 0.330 | 8.77 | 1.74 | 18.6 | 11.0 | 99.1 |
| MonST3R* | 0.295 | 0.240 | 0.313 | 6.49 | 1.78 | 20.3 | 13.4 | 102.5 |
| St4RTrack | 0.278 | 0.247 | 0.370 | 9.37 | 1.76 | 27.2 | 14.6 | 22.5 |
| St4RTrack* | 0.264 | 0.232 | 0.355 | 6.13 | 1.60 | 28.1 | 14.9 | 21.7 |
| POMATO | 0.272 | 0.254 | 0.308 | 6.78 | 1.44 | 26.0 | 15.4 | 81.8 |
| POMATO* | 0.270 | 0.238 | 0.303 | 5.71 | 1.29 | 28.8 | 17.2 | 80.8 |
| Easi3R | 0.308 | 0.302 | 0.324 | 5.15 | 1.55 | 24.7 | 15.2 | 130.9 |
| Trace Anything | **0.234** | **0.218** | **0.295** | **5.09** | **1.06** | **34.5** | 20.6 | **2.3** |

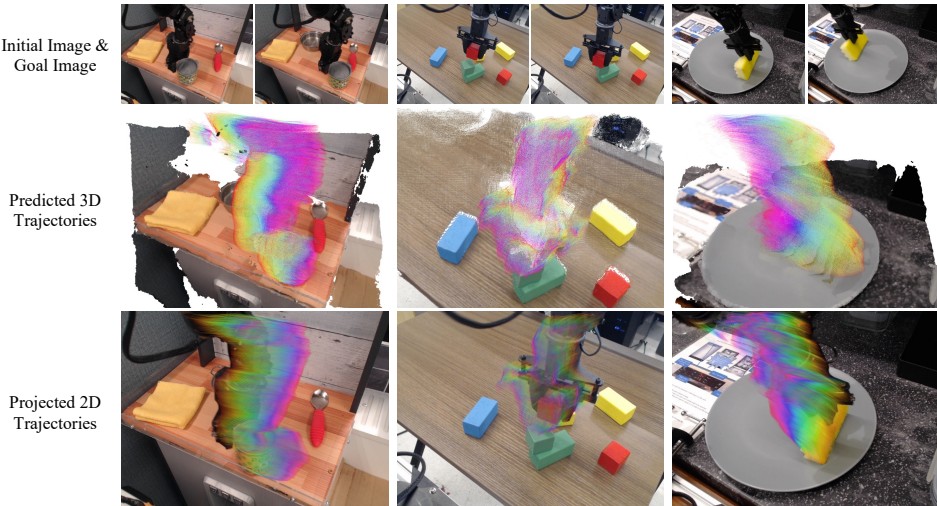

Figure 5: **Image-pair-based trajectory field estimation (goal-conditioned manipulation) on Bridge (Walke et al., 2023).** Given an initial and a goal image, Trace Anything predicts a trajectory field that interpolates the 3D motion of both the robot arm and manipulated objects. We further show the projected 2D trajectories (see Section E.3 and Figure J for details).

## 5.2 QUANTITATIVE EVALUATION

We quantitatively evaluate trajectory field estimation on the *Trace Anything benchmark*, introduced in Section 4. In contrast to established point tracking benchmarks, which evaluate trajectories only from the first frame, our protocol requires *all-to-all* predictions: every pixel in every frame must be associated with a complete trajectory spanning the entire sequence. Evaluation is conducted in two settings: (i) *video-based inference*, where models process 30-frame video clips, and (ii) *image-pair-based inference*, where trajectories are estimated from two frames sampled 5 frames apart. We present other quantitative results in Section E.5 and ablation study in Section E.6.

**Metrics.** We evaluate reconstruction accuracy using end-point error (EPE). Specifically, $\text{EPE}_{\text{mix}}$, $\text{EPE}_{\text{sta}}$, and $\text{EPE}_{\text{dyn}}$ measure the average 3D end-point error over all points, static points, and dynamic points, respectively. To further verify whether the conditions C1 and C2 introduced in Section 3 are satisfied, we introduce two complementary metrics. *Static Degeneracy Deviation (SDD)* quantifies the temporal jitter of trajectories in static regions, where smaller values indicate better compliance with C1. *Correspondence Agreement (CA)* measures how consistently dynamic trajectories are predicted from corresponding pixels in different source frames, with lower values indicating better compliance with C2. For video input, we also report 3D tracking metrics: $\text{APD}_{3D}$ (Average Percentage of Points within a threshold), which measures spatial accuracy, and AJ (Average Jaccard), which captures both spatial alignment and occlusion correctness.

**Baselines.** For video-based inference, we compare against state-of-the-art dynamic scene reconstruction and point tracking approaches, including CoTracker3 (Karaev et al., 2024a) (lifted to

Table 2: **Quantitative results on image-pair-based trajectory field estimation.** CA is reported in $10^{-2}$ and SDD in $10^{-3}$. Best in **bold**, second-best underlined.

| Method | $\text{EPE}_{\text{mix}} \downarrow$ | $\text{EPE}_{\text{sta}} \downarrow$ | $\text{EPE}_{\text{dyn}} \downarrow$ | CA $\downarrow$ | SDD $\downarrow$ | Runtime (s) $\downarrow$ |
|---|---|---|---|---|---|---|
| SEA-RAFT + VGGT | 0.226 | 0.193 | 0.427 | 18.22 | 0.77 | 1.91 |
| RAFT-3D | 0.281 | 0.219 | 0.324 | 17.50 | 0.98 | 0.37 |
| MASt3R | 0.220 | 0.181 | 0.328 | 33.99 | 1.70 | 2.39 |
| MonST3R | 0.206 | 0.167 | 0.346 | 20.10 | 1.25 | 2.51 |
| MonST3R* | 0.198 | 0.155 | 0.321 | 18.10 | 1.27 | 2.44 |
| St4RTrack | 0.211 | 0.202 | 0.325 | 15.33 | 0.63 | 1.39 |
| St4RTrack* | 0.203 | 0.177 | 0.318 | 13.49 | 0.64 | 1.41 |
| POMATO | 0.181 | 0.137 | 0.320 | 19.58 | 0.84 | 4.75 |
| POMATO* | 0.175 | 0.125 | 0.313 | 17.72 | 0.66 | 4.20 |
| Easi3R | 0.284 | 0.269 | 0.323 | 20.41 | 0.91 | 5.08 |
| Trace Anything | **0.135** | **0.106** | **0.304** | **12.41** | **0.54** | **0.20** |

Table 3: **Ablation study on loss terms.** CA is reported in $10^{-2}$ and SDD in $10^{-3}$. Best in **bold**.

| Method | $\text{EPE}_{\text{mix}} \downarrow$ | $\text{EPE}_{\text{sta}} \downarrow$ | $\text{EPE}_{\text{dyn}} \downarrow$ | CA $\downarrow$ | SDD $\downarrow$ |
|---|---|---|---|---|---|
| w/o $\mathcal{L}_{\text{static}}$ | 0.305 | 0.273 | 0.334 | 8.52 | 1.65 |
| w/o $\mathcal{L}_{\text{rigid}}$ | 0.247 | 0.236 | 0.321 | 6.22 | 1.13 |
| w/o $\mathcal{L}_{\text{corr}}$ | 0.241 | 0.220 | 0.303 | 6.17 | 1.10 |
| **Full loss** | **0.234** | **0.218** | **0.295** | **5.09** | **1.06** |

3D using VGGT Wang et al. (2025a)), DELTA (Ngo et al., 2025), SpaTrackerV2 (Xiao et al., 2025b), MonsT3R (Zhang et al., 2025a), St4RTrack (Feng et al., 2025), POMATO (Zhang et al., 2025b) and Easi3R (Chen et al., 2025). For image-pair inference, we compare against the optical flow method SEA-RAFT (Wang et al., 2024b) (lifted to 3D with VGGT Wang et al. (2025a)), the scene flow method RAFT-3D (Teed & Deng, 2021), and several 3D reconstruction approaches, including MASt3R (Leroy et al., 2024), MonST3R (Zhang et al., 2025a), St4RTrack (Feng et al., 2025), POMATO (Zhang et al., 2025b) and Easi3R (Chen et al., 2025). We fine-tune SpaTrackerV2, MonST3R, St4RTrack, and POMATO using the same training scheme as our method( Section D) on Kubric and Trace Anything datasets. These fine-tuned variants are denoted with an asterisk (*).

**Results.** Quantitative results are shown in Tables 1 and 2. Trace Anything achieves the best performance across all metrics, substantially reducing end-point errors in both static and dynamic regions while also attaining the lowest SDD and CA, indicating stronger compliance with consistency conditions. Moreover, it runs over an order of magnitude faster than optimization-based approaches, underscoring the advantage of our one-pass design.

**Runtime breakdown.** As shown in Figure 6, our approach exhibits a total runtime that scales approximately linearly with the number of frames. The fusion transformer is the most time-consuming stage, followed by image encoding and curve evaluation. With single-pass inference and no per-scene optimization or external estimators, it exhibits a clear efficiency advantage, as shown in Tables 1 and 2.

### 5.3 LOSS ABLATION SDUDY

As shown in Table 3, all three loss terms contribute to the final performance. Removing the static regularizer $\mathcal{L}_{\text{static}}$ leads to the largest degradation, especially on static regions, confirming its importance for enforcing no motion in static areas. Dropping the rigidity loss $\mathcal{L}_{\text{rigid}}$ or the correspondence loss $\mathcal{L}_{\text{corr}}$ also degrades performance across all metrics. The full loss configuration achieves the best overall results, indicating that these regularizers are jointly necessary for stable and accurate trajectory estimation.

### 5.4 FACILITATED CAPABILITIES

The Trajectory Field representation and the Trace Anything model facilitate a number of additional capabilities beyond basic tracking.

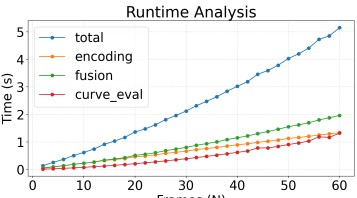

Figure 6: Stage-wise runtime vs. number of input frames.

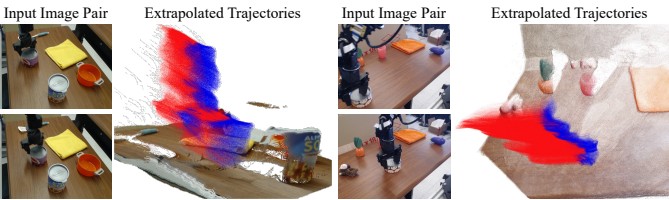

Figure 7: **Velocity-based forecasting.** Per-pixel trajectories are extrapolated by tangent continuation, with reconstructed trajectories in red and extrapolated ones in blue.

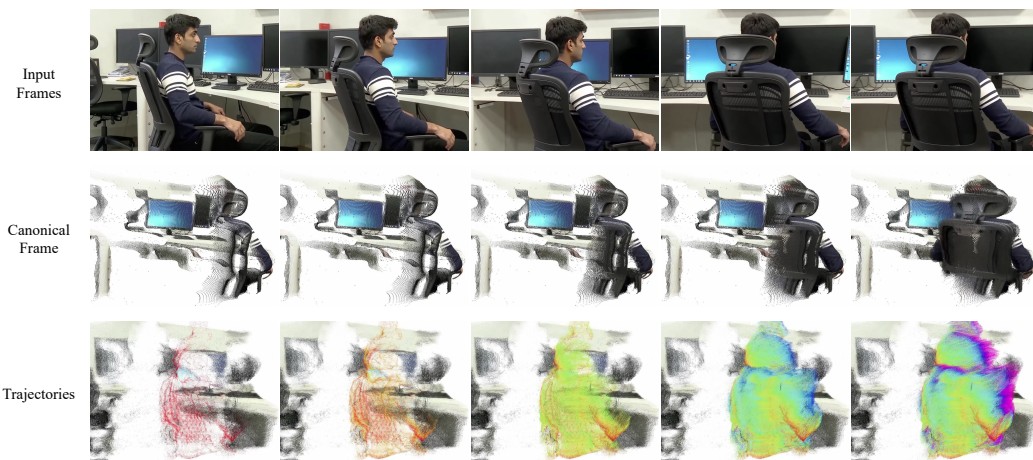

Figure 8: **Spatio-temporal fusion.** The trajectory field can be leveraged to fuse observations of the dynamic entity across different frames into a canonical frame.

**Motion forecasting.** The trajectory field inherently encodes 3D point velocities, enabling *velocity-based forecasting*. As shown in Figure 7, per-pixel trajectories can be extrapolated by tangent continuation, allowing dense motion forecasting without additional predictors. We additionally present *instruction-based forecasting* results in Section E.7.

**Spatio-temporal fusion.** In Figure 8, predicted trajectory fields enable dynamic entities observed across multiple frames to be consistently fused back into a common canonical frame. This provides a mechanism for aggregating partial observations over time, overcoming occlusions and view changes by aligning pixels to a common reference.

## 6 CONCLUSION

We introduced **Trajectory Fields**, a 4D representation that encodes each pixel of each frame as a 3D trajectory, and **Trace Anything**, a feed-forward model that predicts trajectory fields from input frames, eliminating auxiliary estimators and per-scene optimization. To support large-scale learning and evaluation, we developed a synthetic data platform. Experiments show that Trace Anything delivers competitive accuracy and inference efficiency, while enabling additional useful capabilities.

## ACKNOWLEDGEMENTS

This work was done during the first author's ByteDance Seed internship, and supported in part by Research Grant Council of the Hong Kong SAR under Theme-based Research Scheme, grant no. T22-606/23R.

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

# APPENDIX

## CONTENTS

## A  FIELDS

A *field* is a mapping defined on a domain $M$, either a continuous space (e.g., $\mathbb{R}^n$) or a discrete space (e.g., $\mathbb{Z}^n$), to a codomain $V$, which may be a scalar space (e.g., $\mathbb{R}$), a vector space (e.g., $\mathbb{R}^3$), or a function space (e.g., $C^\infty(N)$). Formally, the field is given by

$$\mathcal{F} : M \to V. \tag{15}$$

For instance, the radiance field introduced in (Mildenhall et al., 2020) maps a 3D coordinate $\mathbf{x} \in \mathbb{R}^3$ and a viewing direction $\mathbf{d} \in S^2$ (the unit sphere) to a density $\sigma \in \mathbb{R}^+$ and a color $\mathbf{c} \in \mathbb{R}^3$. This is expressed as

$$\mathcal{R} : \mathbb{R}^3 \times S^2 \to \mathbb{R}^+ \times \mathbb{R}^3,$$
$$(\mathbf{x}, \mathbf{d}) \mapsto (\sigma, \mathbf{c}). \tag{16}$$

As discussed in Section 3.1, the trajectory field introduced in this work is defined as

$$\mathcal{T} : [N] \times [H] \times [W] \to C([0,1], \mathbb{R}^3),$$
$$(i, u, v) \mapsto \mathbf{x}_{i,u,v}(\cdot), \tag{17}$$

where $[N]$, $[H]$, and $[W]$ denote the discrete sets of frame indices and pixel coordinates, respectively, and $\mathbf{x}_{i,u,v} : [0,1] \to \mathbb{R}^3$ is a continuous 3D trajectory for pixel $(u, v)$ in frame $I_i$. The domain is $M = [N] \times [H] \times [W]$, and the codomain is $V = C([0,1], \mathbb{R}^3)$, the space of continuous functions from $[0,1]$ to $\mathbb{R}^3$.

## B  PARAMETRIC CURVES

Our trajectory field representation assigns each pixel to a 3D trajectory, expressed as a parametric curve. In computer graphics, parametric curves are essential for modeling smooth trajectories and surfaces in applications like geometric design and animation (Farin, 2002; Piegl & Tiller, 1997). A spline-based parametric curve $\mathbf{x}(t) : [0,1] \to \mathbb{R}^3$ maps a parameter $t \in [0,1]$ to 3D space, defined as

$$\mathbf{x}(t) = \sum_{k=0}^{n-1} \mathbf{P}_k \phi_k(t), \tag{18}$$

where $\mathbf{P}_k \in \mathbb{R}^3$ are control points and $\phi_k(t)$ are basis functions.

As a widely used class, Bézier curves use Bernstein polynomials as basis functions. A Bézier curve of degree $d$ with $d+1$ control points $\mathbf{P}_0, \mathbf{P}_1, \ldots, \mathbf{P}_d \in \mathbb{R}^3$ is defined as

$$\mathbf{x}(t) = \sum_{i=0}^{d} \mathbf{P}_i B_{i,d}(t), \quad B_{i,d}(t) = \binom{d}{i} t^i (1-t)^{d-i}, \tag{19}$$

where $B_{i,d}(t)$ are Bernstein polynomials (Farin, 2002). Bézier curves interpolate the first and last control points but lack local control, as adjusting one control point affects the entire curve.

B-spline curves, in contrast, provide local control through a knot vector that defines the parameter intervals where basis functions are active. A B-spline curve of degree $p$ with control points $\mathbf{P}_0, \mathbf{P}_1, \ldots, \mathbf{P}_{n-1} \in \mathbb{R}^3$ is defined as

$$\mathbf{x}(t) = \sum_{i=0}^{n-1} \mathbf{P}_i N_{i,p}(t), \tag{20}$$

where $N_{i,p}(t)$ are B-spline basis functions determined by a knot vector via the Cox-de Boor recursion formula (de Boor, 1978).

In our implementation of Trace Anything, we employ cubic B-splines ($p = 3$) with clamped, non-uniform knot vectors to parameterize trajectories $\mathbf{x}_{i,u,v}(t)$. Each segment is defined by four control points, corresponding to the cubic degree ($p = 3$). A trajectory is defined as

$$\mathbf{x}_{i,u,v}(t) = \sum_{k=0}^{n-1} \mathbf{P}_{i,u,v}^{(k)} N_{k,3}(t), \tag{21}$$

where $\mathbf{P}_{i,u,v}^{(k)} \in \mathbb{R}^3$ are control points indexed by $i, u, v$, and $N_{k,3}(t)$ are cubic B-spline basis functions determined by a knot vector $\mathbf{t} = [t_0, t_1, \ldots, t_{m-1}]$. The basis functions are computed via the

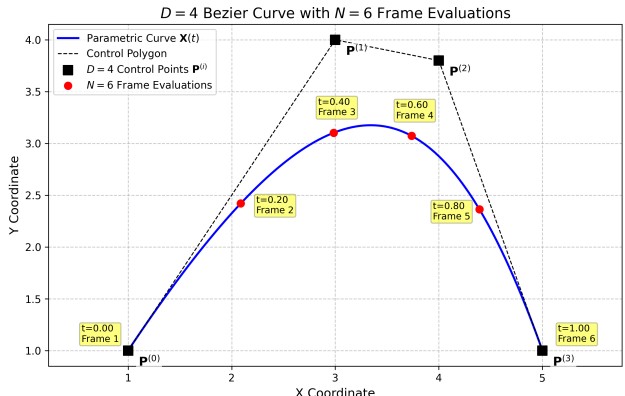

Figure A: **2D Example illustrating $D$ control points and $N$ frame evaluations.** A parametric curve (blue) defined by $D = 4$ control points (black squares) is evaluated at $N = 6$ timestamps corresponding to video frames (red dots).

Cox-de Boor recursion formula:

$$N_{k,0}(t) = \begin{cases} 1 & \text{if } t_k \leq t < t_{k+1} \text{ for } k < n + p, \\ 1 & \text{if } t_k \leq t \leq t_{k+1} \text{ for } k = n + p, \\ 0 & \text{otherwise,} \end{cases} \tag{22}$$

$$N_{k,p}(t) = \frac{t - t_k}{t_{k+p} - t_k} N_{k,p-1}(t) + \frac{t_{k+p+1} - t}{t_{k+p+1} - t_{k+1}} N_{k+1,p-1}(t), \tag{23}$$

for $p = 1, 2, 3$, with non-zero denominators assumed. For $n = 4, 7, 10$ control points, we define knot vectors with multiplicity 4 at $t = 0$ and $t = 1$ to ensure interpolation of the first and last control points ($\mathbf{x}_{i,u,v}(0) = \mathbf{P}_{i,u,v}^{(0)}$, $\mathbf{x}_{i,u,v}(1) = \mathbf{P}_{i,u,v}^{(n-1)}$). The knot vectors $\mathbf{t}_n$ are defined as:

$$\mathbf{t}_n = \begin{cases} [0, 0, 0, 0, 1, 1, 1, 1] & \text{if } n = 4, \\ [0, 0, 0, 0, 0.5, 0.5, 0.5, 1, 1, 1, 1] & \text{if } n = 7, \\ [0, 0, 0, 0, 1/3, 1/3, 1/3, 2/3, 2/3, 2/3, 1, 1, 1, 1] & \text{if } n = 10. \end{cases} \tag{24}$$

Internal knots have multiplicity up to 3, ensuring $C^0$-continuity between segments. Knot differences are precomputed for efficient evaluation. Confidence values are interpolated alongside 3D coordinates $\mathbf{P}_{i,u,v}^{(k)}$ using the same basis functions $\phi_k(t) = N_{k,3}(t)$, enabling uncertainty-aware trajectory modeling.

**Interpreting $D$ and $N$.** We provide a simple 2D illustration in Figure A. Here, $D$ denotes the number of control points that define the underlying smooth trajectory curve (e.g., a Bézier or B-spline curve), while $N$ denotes the number of video frames. The variable $t \in [0, 1]$ is a continuous time parameter of the curve, and each of the $N$ frames corresponds to a specific normalized timestamp $t = \frac{i}{N-1}$.

## C    DATA PLATFORM SAMPLES

We illustrate the diversity of our 4D Scene Data Platform in Figure B. The collection spans a wide range of settings and motions, covering diverse indoor and outdoor environments from public asset libraries and procedural generation, articulated human characters and movable objects with both rigid and non-rigid motion, and smooth camera trajectories sampled around active regions to mimic natural filming.

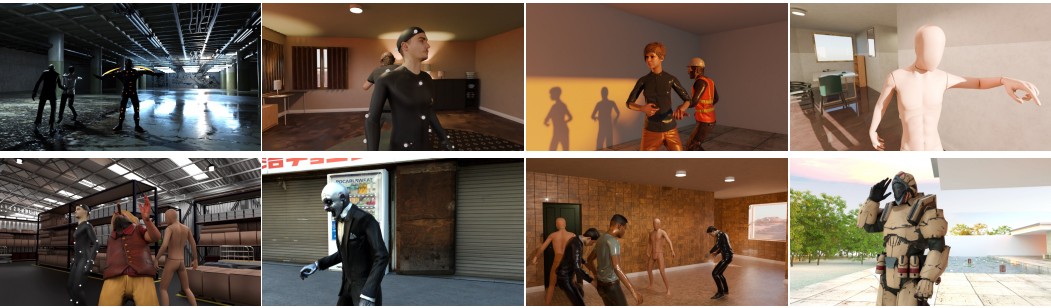

Figure B: Sample renderings from our 4D Scene Data Platform.

## D    EXPERIMENTAL DETAILS

We generate training data using Kubric (Greff et al., 2022) and our proposed 4D scene data engine. Specifically, we render 20K videos with 24 frames each using Kubric, with half containing continuous camera motion and the other half discrete camera motion, and over 10K videos with 120 frames each from our engine. While Kubric equips models with preliminary ability to capture rigid object motion, it is largely limited to rigid dynamics and textured backgrounds. Our engine complements this with diverse non-rigid object motions and more complex, varied environments.

Our released model uses an image encoder and fusion transformer initialized with Fast3R (Yang et al., 2025a) pretrained weights, while the CP heads are randomly initialized. For the choice of parametric curves, our released model adopts B-splines, as detailed in Section B. All models are trained on images at a resolution of 512 pixels on the longest side, using AdamW (Loshchilov et al., 2017) with a learning rate of 0.0001 and a cosine annealing schedule.

In the first stage, we train on 20K Kubric videos; in the second stage, we use a mixture of 20K Kubric videos and 10K from our engine. We adopt a batch size of 1, with each batch sampling up to 30 frames. Training takes 7.22 days on 32 NVIDIA A100 80GB GPUs. To enable efficient large-scale training, we leverage FlashAttention (Dao et al., 2022; Dao, 2023) for improved time and memory efficiency, and apply DeepSpeed ZeRO Stage 2 (Rajbhandari et al., 2020), which partitions optimizer states, moment estimates, and gradients across machines.

**Baseline training datasets.** We present a comparison of the training data used by different methods in  Table A. We group the datasets by label type into Depth or Point Maps and Trajectories, where the latter encompasses supervision from both 3D tracks and scene flows. On Trace Anything benchmark, we also report fine-tuned variants of competitive baselines (MonsT3R, POMATO, St4RTrack, and SpaTrackerV2) on the Kubric and Trace Anything datasets using the same training recipe as our method, designated with * in Table 1 and Table 2.

## E    ADDITIONAL EXPERIMENTAL RESULTS

In this section, we present additional experimental results.  Please also refer to the supplementary materials for video results, including the presented features, interactive visualization demos, and qualitative comparisons.

### E.1    COMPARISON ON REAL-WORLD BENCHMARKS

**KITTI scene flow.** We evaluate scene flow on the KITTI Scene Flow 2015 dataset (Menze et al., 2015), comparing our method with VGGT (Wang et al., 2025a), MASt3R (Leroy et al., 2024), and MonST3R (Zhang et al., 2025a). Because KITTI images have a very wide aspect ratio, which differs from the training distributions of our method and all competing baselines, we fine-tune each approach on the Trace Anything dataset with an aspect ratio of $0.3$ to $0.4$ for 1 epoch. Despite this brief adaptation, KITTI should still be regarded as an out-of-distribution benchmark for our method. As shown in the qualitative results in Figure C, our method accurately identifies moving objects and recovers their motion direction and velocity, even in scenes containing multiple independently

| Method | Depth or Point Map Datasets | Trajectory Datasets |
|---|---|---|
| RAFT-3D | — | FlyingThings3D, KITTI |
| MASt3R | Habitat, ARKitScenes, BlendedMVS, MegaDepth, StaticScenes3D, ScanNet++, CO3D-v2, Waymo, Mapfree, WildRGB, VirtualKITTI, Unreal4K, TartanAir, internal dataset | — |
| VGGT | Co3Dv2, BlendMVS, DL3DV, MegaDepth, Kubric, WildRGB, ScanNet, HyperSim, Mapillary, Habitat, Replica, MVS-Synth, PointOdyssey, VirtualKITTI, Aria Synthetic Environments, Aria Digital Twin, Objaverse-like synthetic assets | — |
| Easi3R | *Training-free (built upon DUSt3R)* | |
| MonST3R | PointOdyssey, TartanAir, Spring, Waymo | — |
| POMATO | Tartanair, ParallelDomain4D, Carla | PointOdyssey, Dynamic Replica |
| SpaTrackerV2 | VKITTI, TartanAir, Spring, DL3DV, BEDLAM, MVS-Synth, CO3Dv2, COP3D, WildRGBD, ScanNet++, OmniObject3D | PointOdyssey, Dynamic Replica, Kubric |
| DELTA | — | Kubric |
| St4RTrack | — | PointOdyssey, Dynamic Replica, Kubric |
| **Ours** | — | Kubric, Trace Anything |

Table A: Training datasets used by different methods.

moving vehicles. We evaluate both *short-range scene flow* (between consecutive image pairs) and *long-range scene flow* (across 15 consecutive frames). The quantitative metrics are: 1) *End-Point Error (EPE)*, defined as the $L_2$ norm between the predicted scene flow and the ground truth, and 2) *Accuracy (Acc.)*, defined as the percentage of points whose relative error is below $10\%$ The quantitative results in Table B show that our method achieves the best performance across all metrics, outperforming prior approaches in both short- and long-range evaluations.

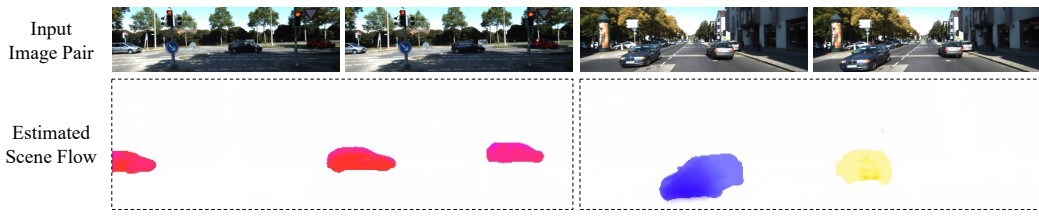

Figure C: Qualitative results of scene flow estimation on KITTI. The $x$ and $z$ flow components are color-coded for visualization.

Table B: Quantitative comparison of scene flow estimation on KITTI. The best results are **bolded** and the second-best are underlined.

| Method | Short-Range | | Long-Range | |
|---|---|---|---|---|
| | EPE ↓ | Acc. ↑ | EPE ↓ | Acc. ↑ |
| VGGT | 0.415 | 0.510 | 0.407 | 0.536 |
| MonST3R | 0.401 | 0.559 | 0.409 | 0.515 |
| MASt3R | 0.425 | 0.487 | 0.438 | 0.481 |
| Ours | **0.375** | **0.574** | **0.372** | **0.589** |

**3D point map estimation.** While our approach is designed and trained for trajectory field estimation, it naturally produces dense 3D point maps as an intermediate representation. We provide both qualitative and quantitative comparisons of these point maps against VGGT (Wang et al., 2025a), MASt3R (Leroy et al., 2024), and MonST3R (Zhang et al., 2025a) on the Bonn dataset (Palazzolo et al., 2019) and TUM Dynamics (Sturm et al., 2012). As shown in Figures D and E, baseline methods frequently exhibit artifacts around moving or non-rigid subjects (this is particularly evident with VGGT and MASt3R). In contrast, our approach consistently preserves object geometry more accurately and maintains sharper motion boundaries, resulting in a more accurate and visually coherent scene reconstruction. The quantitative results in Table C show that our method achieves the best performance across all metrics on the Bonn dataset and performs on par with VGGT on TUM Dynamics.

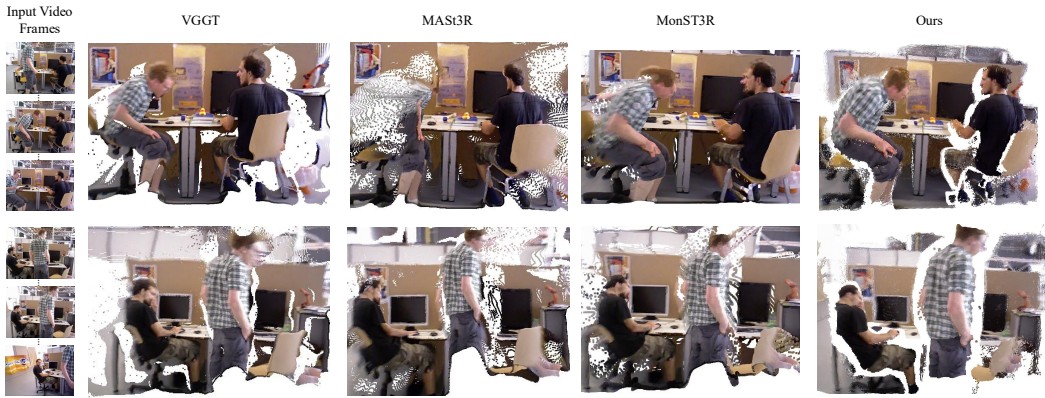

Figure D: Qualitative comparison of 3D pointmap estimation on the TUM Dynamics dataset.

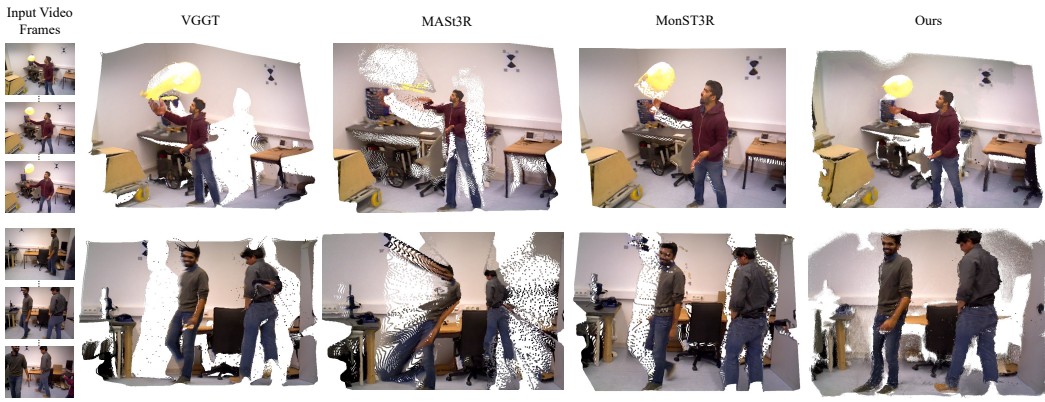

Figure E: Qualitative comparison of 3D pointmap estimation on the Bonn dataset.

Table C: Quantitative comparison of point map estimation on TUM Dynamics and Bonn. The best results are **bolded** and the second-best are underlined.

| Method | TUM Dynamics | | | Bonn | | |
|---|---|---|---|---|---|---|
| | Accuracy ↓ | Completeness ↓ | Overall ↓ | Accuracy ↓ | Completeness ↓ | Overall ↓ |
| VGGT | **0.351** | **0.227** | **0.289** | 0.481 | 0.403 | 0.442 |
| MonST3R | 0.410 | 0.300 | 0.355 | 0.530 | 0.522 | 0.526 |
| MASt3R | 0.553 | 0.390 | 0.471 | 0.577 | 0.416 | 0.497 |
| Ours | 0.381 | 0.247 | 0.314 | **0.363** | **0.270** | **0.317** |

## E.2 ADDITIONAL QUALITATIVE RESULTS

**Videos as input.** Figure F presents the qualitative results of our point map estimation, demonstrating effective handling of complex multi-object motion scenarios compared to baselines. Furthermore, Figure G showcases the superior fidelity of our Trajectory Field estimation. These challenging video inputs highlight the ability of our representation to model the fine structural motion of objects, accurately capturing dynamics in complex geometries such as the articulated limbs of a robot dog or the caribou horns.

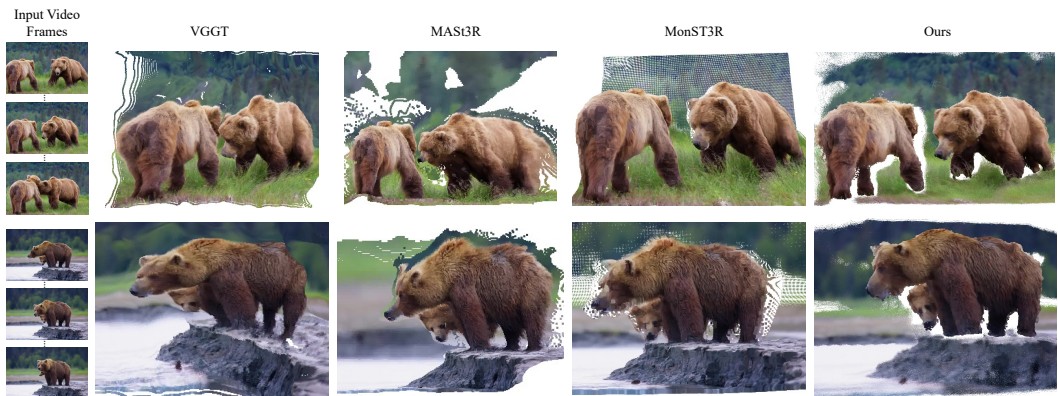

Figure F: Qualitative comparison of 3D pointmap estimation on Internet video clips.

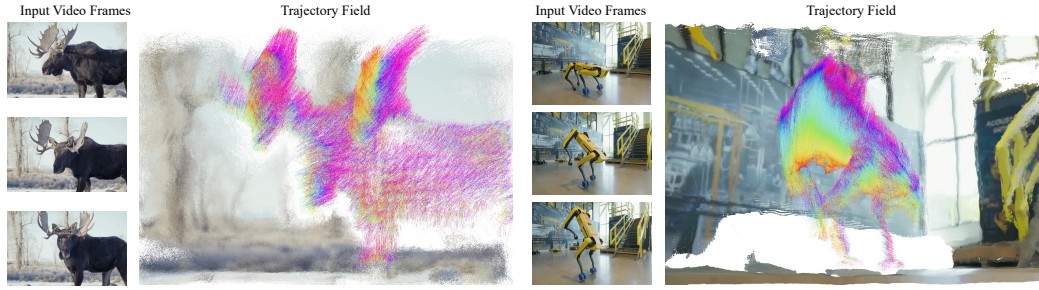

Figure G: Qualitative results of trajectory field estimation on Internet video clips.

**Image pairs as input.** Figure H presents additional qualitative examples using image pairs as input, highlighting the ability of our trajectory field to capture fine-grained, pixel-level 3D motion.

**Unstructured image sets as input.** Beyond videos or image pairs, our method also handles unstructured, unordered image sets, a setting not addressed by prior work. The inputs lack both temporal ordering and continuous camera motion, yet our framework by design can also cope with such challenging cases. As shown in Figure I, our method predicts plausible trajectory fields and camera poses under these conditions. For clarity, we present the input images in chronological order, although no sequence information is provided to the model.

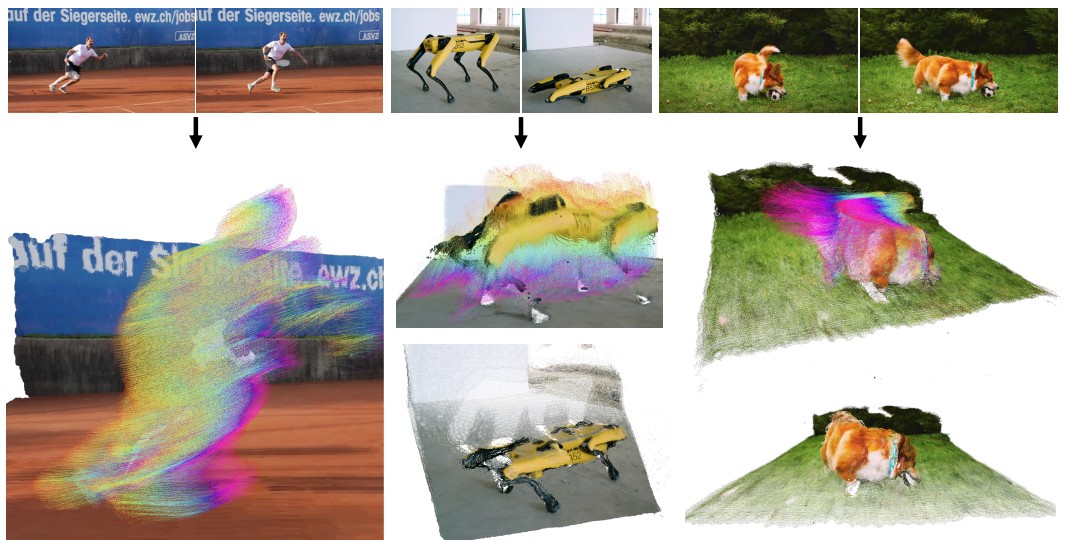

Figure H: Additional qualitative results with image pairs as input.

### E.3 2D Trajectories, Dynamic Masks, Scene Flow, and Camera Poses

The outputs of Trace Anything can naturally yield 2D trajectories, dynamic masks, scene flow, and camera poses.

**2D trajectories.** Given the predicted per-pixel 3D trajectories, and with known or estimated camera parameters, we can project them into the image plane to obtain 2D trajectories. In Figure J, we overlay the projected 2D trajectories on the first input frame. We also demonstrate this feature in Figure 5.

**Dynamic masks.** Our method effectively disentangles static and dynamic components. After Trace Anything predicts control points, we compute the variance over the control-point set associated with each pixel; thresholding this per-pixel variance yields a dynamic mask that cleanly separates static from dynamic regions, as illustrated in Figure K.

**Scene flow.** Given an input image pair, the scene flow can be obtained as the difference between the two endpoints of the predicted trajectories. In Figure L, we present a 4D reconstruction together with the estimated scene flow from an image pair in the *Spring* dataset. To highlight robustness under long-range motion, the two images are chosen from non-consecutive frames.

**Camera poses.** Since Equation (6) provides a world-coordinate point map for each image, we follow Yang et al. (2025a) (Sec. 4.2) to estimate focal length, rotation, and translation. Our method handles both continuous camera motion in videos and discrete poses from unordered image sets. As shown in Figure M, it correctly recovers camera motion even in dynamic scenes—for example, forward camera movement with perpendicular object motion, or objects in free fall captured by an unordered image set. In the second example, we present the input images in chronological order, although no temporal information is provided to the model.

### E.4 Qualitative Comparison

We provide qualitative comparisons of reconstructed point clouds on DAVIS (Perazzi et al., 2016). As shown in Figure N, our method better preserves fine object details (e.g., the elephant's tail and the flamingo's neck), correctly handles complex motion, and disentangles static and dynamic objects. Please refer to the supplementary videos for clearer visual comparisons.

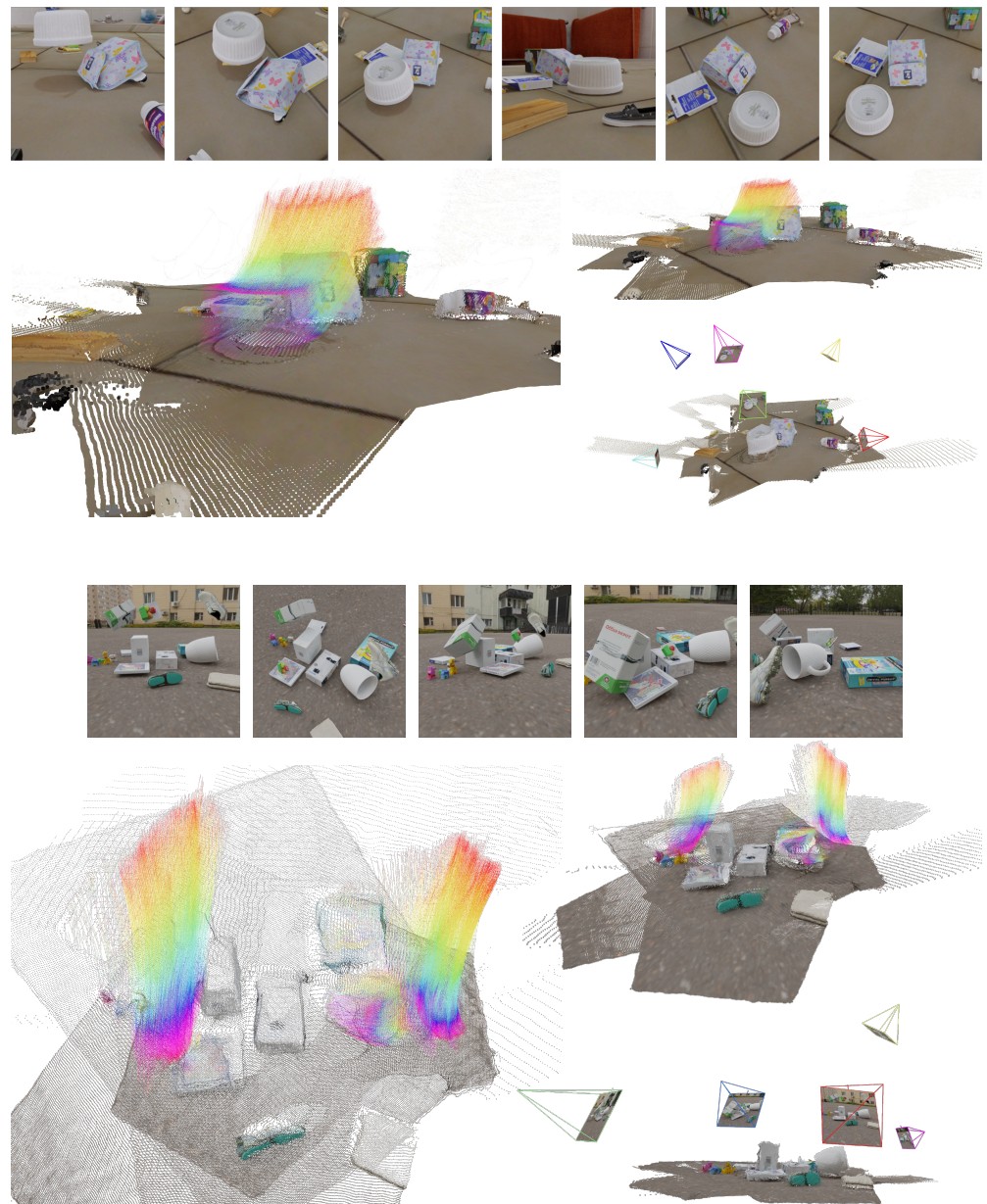

Figure I: Trajectory fields and camera poses estimated from an unstructured, unordered image set. No sequence information is provided to the model.

### E.5 ADDITIONAL QUANTITATIVE COMPARISON

**Out-of-distribution input.** To evaluate our model under out-of-distribution conditions, we construct an additional benchmark from PointOdyssey (Zheng et al., 2023), consisting of 50 videos of 30 frames each. Our model has never been trained or fine-tuned on PointOdyssey. As shown in Table D, our method maintains advantages across all metrics as well as inference efficiency.

**3D tracking.** Although our primary task is trajectory field estimation, our method achieves strong results on 3D tracking without task- or dataset-specific fine-tuning. We quantitatively compare against other approaches on the TAPVid-3D (Koppula et al., 2024) benchmark. For each subset of TAPVid-3D (ADT, DriveTrack, and PStudio), we sample 50 videos of 60 frames each, using ev-

Input Frames          Trajectory Field          Projected 2D Trajectories

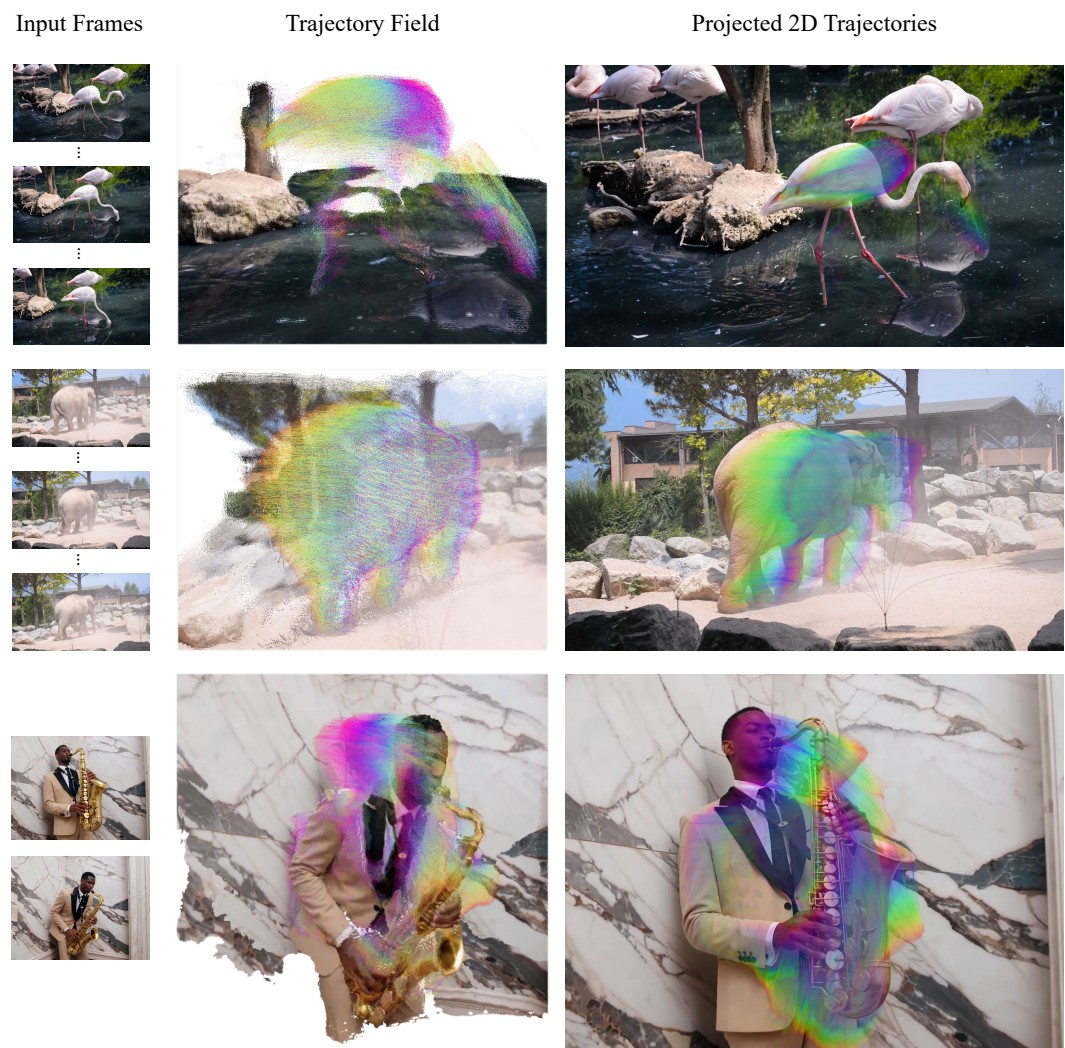

Figure J: Projected 2D trajectories overlaid on the first input frame.

Table D: **Quantitative results on out-of-distribution data.** CA is reported in $10^{-2}$ and SDD in $10^{-3}$. Best in **bold**, second-best underlined.

| Method | $EPE_{mix}$ ↓ | $EPE_{sta}$ ↓ | $EPE_{dyn}$ ↓ | CA ↓ | SDD ↓ | Runtime (s) ↓ |
|---|---|---|---|---|---|---|
| St4RTrack | 0.269 | 0.243 | 0.325 | 9.82 | 1.70 | 19.9 |
| POMATO | 0.344 | 0.319 | 0.397 | 6.24 | 1.72 | 84.1 |
| Easi3R | 0.368 | 0.307 | 0.376 | 7.10 | 1.99 | 125.1 |
| Trace Anything | **0.256** | **0.212** | **0.319** | **6.19** | **1.37** | **2.3** |

ery other frame as input, and report $APD_{3D}$ (average percent of points within a threshold, measuring spatial accuracy) and AJ (average Jaccard, capturing both spatial and occlusion correctness).

In Table E, we present these quantitative results. Notably, SpaTracker (Xiao et al., 2024) is designed and trained for 3D tracking. Our approach remains competitive, surpassing it on some metrics and running orders of magnitude faster, as SpaTracker is limited to a fixed number of query points per run, whereas our model performs per-pixel tracking in a single forward pass.

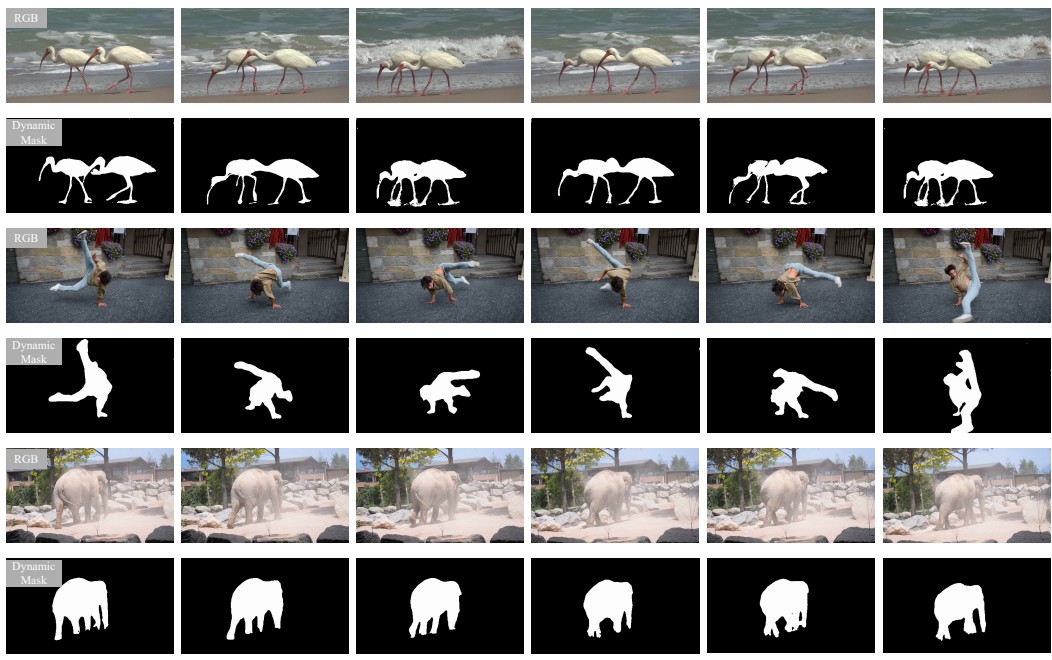

Figure K: Dynamic mask estimation.

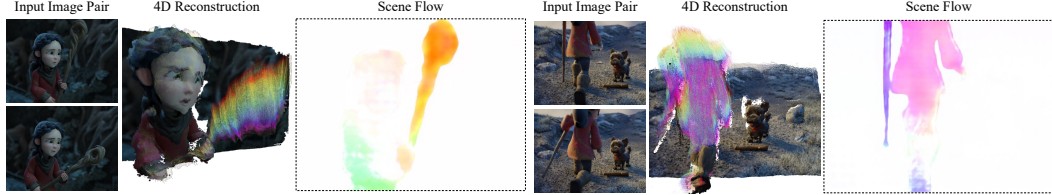

Figure L: **4D reconstruction and scene flow from a single image pair.** From a non-consecutive image pair in the *Spring* dataset, our method recovers both the 4D reconstruction and the scene flow, with $x$ and $z$ components color-coded for visualization.

### E.6    ABLATION STUDY

Table F presents ablation studies on our Trace Anything benchmark, evaluating both the choice of geometric backbone and the type of parametric curve. For the geometric backbone, we compare the effect of initializing the image encoder and fusion transformer with different pretrained models, including Fast3R (Yang et al., 2025a), VGGT (Wang et al., 2025a), and "None" (following the Fast3R architecture but with random initialization). For the parametric curve types, we evaluate polynomial curves* as well as Bézier and B-spline curves with varying numbers of control points.

As shown in Table F, polynomial curves underperform because their parameters lack the clear geometric and physical interpretability. In contrast, B-spline curves with ten control points achieve the best overall performance, and accuracy generally improves as the number of control points increases. For the backbone, training without pretrained initialization struggles to converge. Compared with Fast3R, VGGT yields modest gains on certain metrics but incurs substantially higher runtime. Nonetheless, we observe VGGT can be beneficial in settings that demand fine structural detail or involve large-baseline scenarios. Based on these results, we adopt Fast3R with B-spline curves (10 control points) as the default configuration in Trace Anything.

---

*Although our method restricts parametric curves to control-point–based ones such as Bézier and B-spline, we experimented with polynomial curves during the early development phase.

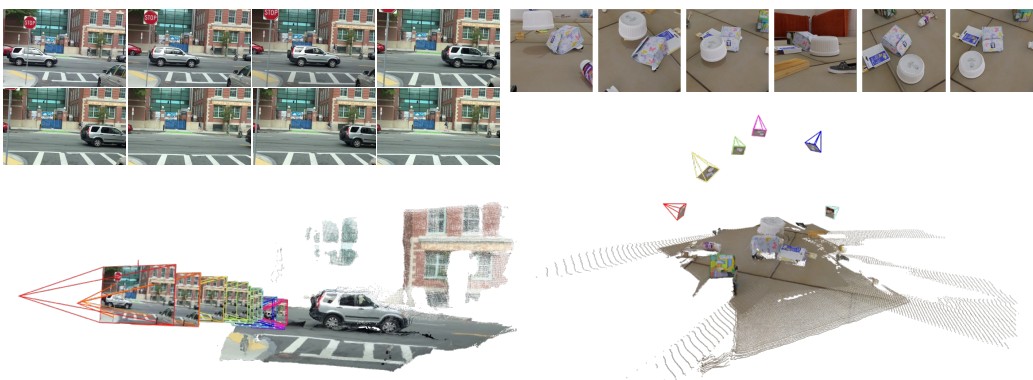

Figure M: Estimated camera poses over the 4D reconstruction.

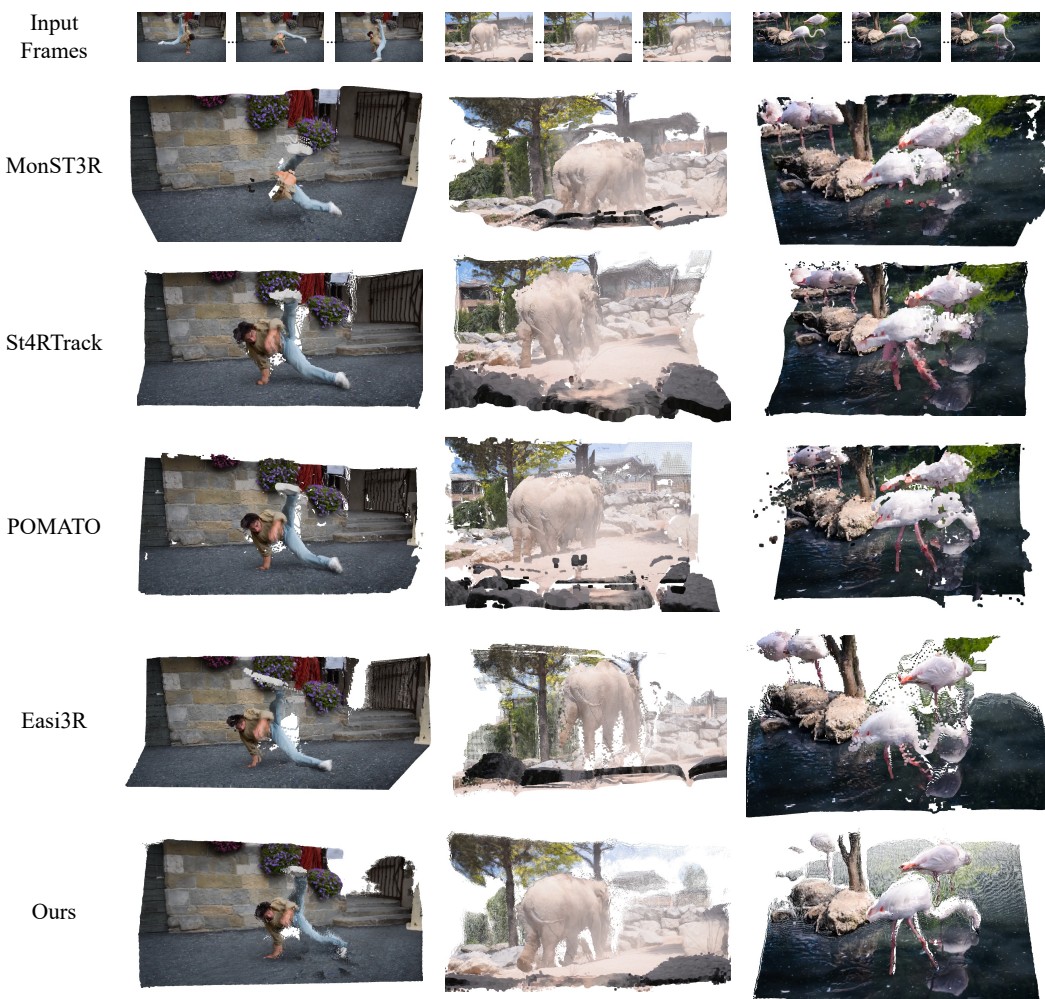

Figure N: **Qualitative comparison on DAVIS (Perazzi et al., 2016).** Our method better recovers fine details and handles complex motion while disentangling static and dynamic objects.

## E.7 FACILITATED CAPABILITY: INSTRUCTION-BASED FORECASTING

As an extension of Section 5.4, our method also enables instruction-based motion forecasting. With natural language instructions as input, we leverage image or video generation models to produce

Table E: **Quantitative results on 3D tracking.** Best in **bold**, second-best underlined.

| Method | ADT | | DriveTrack | | PStudio | | |
|---|---|---|---|---|---|---|---|
| | $APD_{3D}$ ↑ | AJ↑ | $APD_{3D}$ ↑ | AJ↑ | $APD_{3D}$ ↑ | AJ↑ | Runtime (s)↓ |
| VGGT + CoTracker | 8.9 | 9.7 | 6.2 | 5.4 | 8.6 | 5.8 | 172.4 |
| St4RTrack | 15.2 | 13.4 | 8.5 | 7.4 | 7.2 | 6.9 | 18.9 |
| POMATO | 18.2 | 13.6 | 11.3 | 7.8 | 12.2 | 8.3 | 69.2 |
| SpaTracker | 18.3 | **17.4** | **16.0** | **10.1** | 16.2 | 10.3 | 191.1 |
| Trace Anything | **20.5** | 15.6 | 15.5 | 9.6 | **16.3** | **10.8** | **2.1** |

Table F: **Ablation study on Trace Anything benchmark.** CA is reported in $10^{-2}$ and SDD in $10^{-3}$. Best in **bold**, second-best underlined.

| Backbone | Curve Type | $EPE_{mix}$ ↓ | $EPE_{sta}$ ↓ | $EPE_{dyn}$ ↓ | CA ↓ | SDD ↓ | Runtime (s)↓ |
|---|---|---|---|---|---|---|---|
| None | B-Spline (10 control points) | 0.472 | 0.416 | 0.505 | 8.17 | 1.08 | 2.3 |
| Fast3R | Polynomial (degree 3) | 0.619 | 0.582 | 0.673 | 9.19 | 1.10 | 1.8 |
| Fast3R | Bezier (4 control points) | 0.299 | 0.271 | 0.312 | **5.08** | 1.11 | **1.7** |
| Fast3R | Bezier (10 control points) | 0.238 | 0.224 | 0.319 | 6.13 | 1.08 | 2.5 |
| Fast3R | B-Spline (4 control points) | 0.281 | 0.264 | 0.330 | 6.01 | 1.08 | **1.7** |
| Fast3R | B-Spline (7 control points) | 0.237 | 0.229 | 0.317 | 5.81 | 1.11 | 2.1 |
| Fast3R | B-Spline (10 control points) | **0.234** | **0.218** | **0.295** | 5.09 | **1.06** | 2.3 |
| VGGT | B-Spline (10 control points) | 0.236 | 0.221 | 0.276 | 6.11 | 1.07 | 7.2 |

future states conditioned on the instructions, and then apply Trace Anything to lift these forecasts into 2D trajectory fields.

In Figure O, we forecast hand–object interactions under different instructions. We use Gemini 2.5 Flash Image (Nano Banana) to generate images of future states corresponding to each instruction, and then apply Trace Anything on the generated image pairs to estimate the trajectory field.

In Figure P, we forecast robot actions conditioned on different instructions. We use Seedance 1.0 (Gao et al., 2025) to generate videos of future states for different instructions, and then apply Trace Anything to predict the trajectory fields from the generated videos.

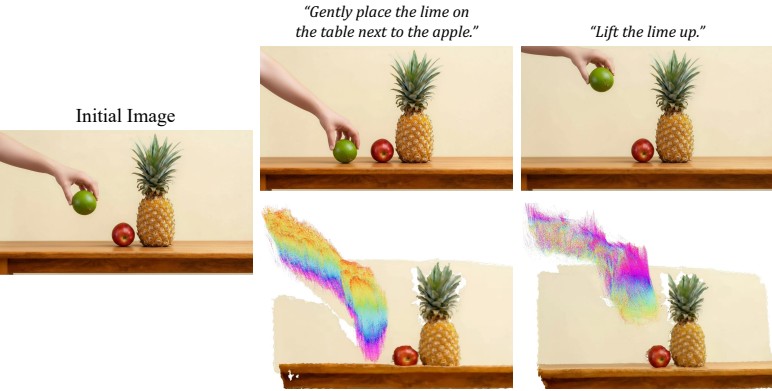

Figure O: Instruction-based forecasting. Future states are generated using Nano Banana.

### E.8 ADDITIONAL ANALYSES

**Temporal interpolation.** To evaluate the model's ability to interpolate motion over continuous time, we conduct an experiment where each input consists of 6 frames at time steps $t = \{0, 0.2, 0.4, 0.6, 0.8, 1.0\}$. We estimate the trajectory field and then evaluate the predicted trajectories both at these given frames and at the unseen interpolated time steps $t = \{0.1, 0.3, 0.5, 0.7, 0.9\}$. This evaluation is performed over 100 sets of input frames. As shown in Figure Q, the EPE at inter-

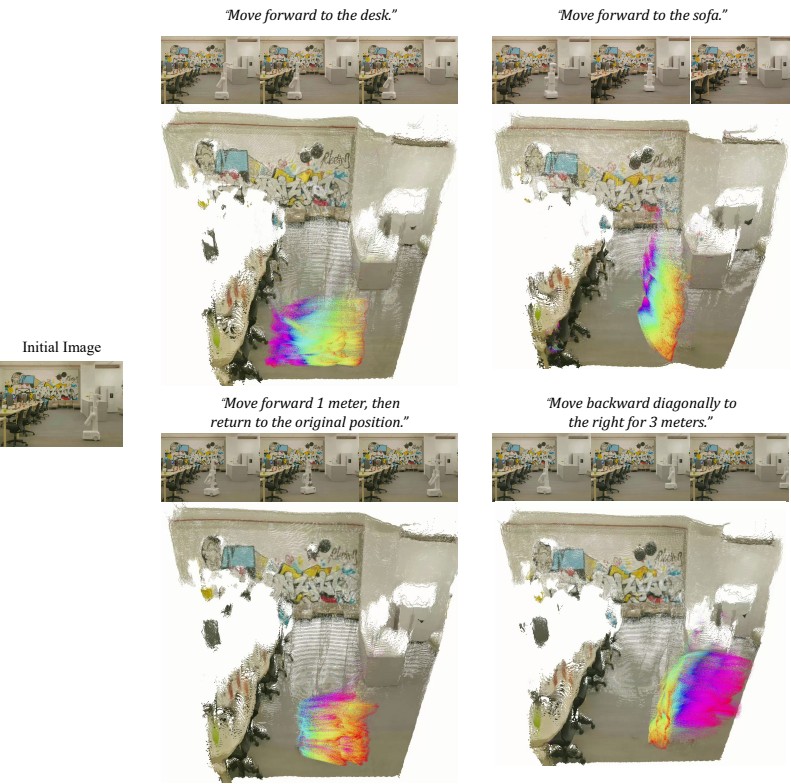

Figure P: Instruction-based forecasting. Future states are generated using Seedance 1.0.

polated time steps is slightly higher than the EPE at the given frames. This mild increase is expected and indicates that the model tends to perform marginally better at the original given frames.

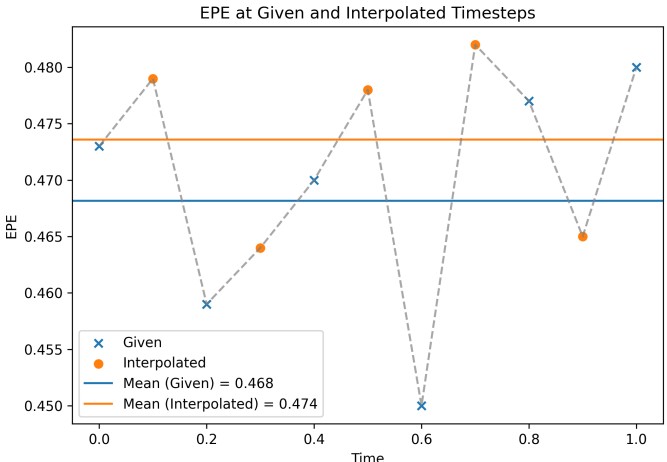

Figure Q: **EPE at given and interpolated timesteps.** EPE at interpolated time steps is slightly higher than at the given frames.

**Curve capacity analysis.** To validate the choice of parametric curve, we compare different spline-based models—linear (piecewise-linear), Bézier, and B-spline with varying numbers of control points, and additionally compare against the network prediction of our approach and the state-of-

the-art 3D tracker SpaTrackerV2 (Xiao et al., 2025b). As shown in Table G, a cubic B-spline with $D=10$ control points is sufficiently expressive to model long and complex 3D trajectories. When optimally fit to all trajectories from 20 challenging real-world scenes in TAPVid-3D (Koppula et al., 2024) (each spanning 120 frames), the $D=10$ B-spline achieves lower fitting error than SpaTrackerV2 (Xiao et al., 2025b). This confirms that the spline representation itself is not the bottleneck, and its capacity is sufficient for the vast majority of real-world motion.

Table G: **Curve-fitting accuracy.** Optimal fitting of various parametric curves (linear, Bézier, B-spline) to complex real-world 3D trajectories from TAPVid-3D (Koppula et al., 2024). A cubic B-spline with $D=10$ control points achieves the lowest error, outperforming state-of-the-art 3D tracking methods, indicating that curve capacity is not the limiting factor.

| Methodology | Curve Type | # Control Points | RMSE | Max Error |
|---|---|---|---|---|
| **Optimization-Based Fitting** | Linear | 2 | 1.530 | 2.180 |
| | Linear | 4 | 1.251 | 2.053 |
| | Linear | 7 | 0.481 | 0.838 |
| | Linear | 10 | 0.311 | 0.732 |
| | Bezier | 4 | 1.029 | 1.724 |
| | Bezier | 7 | 0.482 | 1.208 |
| | Bezier | 10 | 0.225 | 0.474 |
| | B-spline | 4 | 1.029 | 1.724 |
| | B-spline | 7 | 0.294 | 0.639 |
| | B-spline | 10 | **0.200** | **0.444** |
| **SpaTrackerV2** | – | – | 0.592 | 1.270 |
| **Ours** | B-spline | 10 | 0.520 | 1.157 |

## F  LIMITATIONS

Since Trace Anything is trained for trajectory field estimation, we rely on synthetic data to obtain dense annotations. This inevitably introduces a domain gap with real-world scenarios. Incorporating partial annotations from real data may help bridge this gap and represents a promising direction for future work.

Our parametric curve representation, with a limited number of control points, has restricted expressive power for highly complex motions. In such cases, we mitigate the issue by clipping trajectories into fixed window sizes or downsampling frames. However, these strategies may fail in scenarios such as repeated back-and-forth motion, and performance also degrades as the number of frames increases. A more fundamental solution likely requires training with larger-scale datasets with high quality.

As the first attempt at dense per-pixel trajectory field estimation, our approach offers efficiency advantages but may be less precise than sparse 3D tracking methods (Xiao et al., 2024; 2025b). Incorporating fine-grained point-level estimation from such methods into our framework could be an interesting direction for future research.

## G  LLM USAGE DECLARATIONS

We declare that Large Language Models (LLMs) were used in a limited capacity during the preparation of this manuscript. Specifically, LLMs were employed for grammar checking, word choice refinement, and typo correction. All core technical contributions, experimental design, analysis, and conclusions are entirely our own. The use of LLMs did not influence the scientific methodology, result interpretation, or theoretical contributions of this research.

