# OpenReview forum: "Trace Anything: Representing Any Video in 4D via Trajectory Fields"
_ICLR.cc/2026/Conference — ICLR 2026 Poster_

### Official Review · Reviewer_o94T · 2025-10-15

**Soundness:** 4
**Presentation:** 4
**Contribution:** 4
**Rating:** 10
**Confidence:** 4

**Summary:**

This paper introduces a feed-forward method to directly represent the dynamic video in a continuous trajectory field, which can predict the 2d trajectories (2d in a way it predicts for the image coordinate) for the image sequence at any queried time frame. Specifically, this method uses B-splines as basis functions and uses the predicted feature-based “points” to get the final trajectory. The “control points” are predicted using the large-model image encoder. The authors have shown good performance on video/image pairs to the trajectory dataset DAVIS and BridgeData V2, along with other applications including motion forecasting, spatio-temporal fusion, etc, showing the great potential of this proposed trajectory field representation.

**Strengths:**

- To directly represent the image sequence/video as a continuous trajectory field is reasonable and has many advantages, such as no need for depth/geometry/camera priors, can directly query the motions, etc.
- This continuous trajectory field representation has many practical applications, as shown in the paper. For example, the goal-conditioned trajectory generation for robotic manipulation. This could be a great future work.
- As a direct feed-forward network, the computation is efficient and the model is fast during inference.
- The authors have also released (will release) a benchmark dataset for the trajectory estimation, which is needed from the research community.
- The paper is well-written, and all the figures and included videos are of high quality.

**Weaknesses:**

- The authors could provide more experimental results on video to trajectory datasets to further highlight its robustness and generalizability.
- I am a bit confused by the definition in the paper referring to 3d points. It seemed that the predicted “3d points” in the paper referred to the (u,v) coordinate at a specific time frame t. It would be better to make this notation clearer.

**Questions:**

- Can the model handle large camera motions? I found that in the
- Could the authors provide an intuitive explanation for the condition 2?
- I wondered if the authors have trained their image encoder end-to-end for the experiments they showed in the paper or used a pretrained model here.
- What about the generalizability of the proposed method? I have seen one experiment in the appendix, how about more different OOD data? Also, does this generalizability come from the large-model image decoder or the continuous trajectory representation itself?

---

> ### Author Response · Authors · 2025-11-25
> **Response to Reviewer o94T**
>
> We thank the reviewer for the encouraging assessment. We appreciate the thoughtful comments and address the questions and suggestions below.
>
> **More results on video-based estimation (W1).** In the revision, we include more video-input results in **Appendix E.1** (real-world benchmarks) and **Appendix E.2** (additional qualitative results). These examples further demonstrate robustness and generalizability to diverse real-world video scenarios across tasks such as trajectory fields, point maps, and scene flow estimation.
>
> **Clarification on “3D points” (W2).** As defined in Equation 1, a pixel $(u,v)$ in frame $i$ is mapped to a 3D trajectory, whose evaluation at time $t$ yields a 3D point on the corresponding point map. We have improved **Section 3.1** and **Appendix B** to clarify this notation.
>
> **Large camera motion (Q1).** Because our model represents dynamic scenes in a world frame rather than in the camera frame, it naturally handles large viewpoint changes. As shown in **Figure M**, the model remains stable even under large or teleportation-like camera motions.
>
> **Intuition for Condition 2 (Q2).** Condition 2 states that if two pixels from different frames observe the **same physical 3D point** in the scene, then the model should assign them the **same underlying 3D trajectory**. In other words, *the identity of a physical point should not change over time.* If a pixel in frame $i$ and a pixel in frame $j$ correspond to the same real-world point, both should lie on the same 3D curve traced by that point’s motion. This ensures temporal and geometric consistency across frames.
>
> **Image Encoder (Q3).** We initialize the image encoder using the pretrained backbone from Fast3R/VGGT, and then fine-tune it jointly with all other modules. The revised manuscript states this clearly in **Appendix D**.
>
> **Generalizability (Q4).** In the revision, we include additional quantitative and qualitative OOD results in **Appendix E.1** and **Appendix E.2**, demonstrating that the method generalizes well across diverse real-world scenarios. This includes driving scenes from KITTI and challenging robot- and animal-motion videos in **Figures C, F, and G**, all very different from the human-motion data used for training. Generalization arises from both components of the system: the pretrained backbone contributes strong spatial modeling capability (primarily for static scenes), while the trajectory field representation enables robust motion estimation even when the observed motions differ substantially from those in the training domain.

---

> > ### Comment · Reviewer_o94T · 2025-11-27
> > **Response to the authors**
> >
> > Thanks for the detailed explanations.
> >
> > I appreciate the authors for providing more video-based estimation results, the clarifications on some arguments in the paper, and some additional results to back up for the generalizability of the proposed method. I think the explanation for condition 2 is quite clear, and this could be added to make it more specific---instead of directly saying "mapping to the same 3d trajectory", claiming that they correspond to the same 3d points, therefore, having the same 3d trajectory.
> >
> > One point I want to mention is the large camera motion. Although it makes scenes that the model represents "dynamic scenes in a world frame rather than in the camera frame", the motion can still be rather smooth and sparse in the settings. I wondered if more quantitative results could be provided to further strengthen the paper.
> >
> > Nevertheless, I still think this is a novel and reasonable direct forward method for representing the video as the trajectory field. The idea is neat and effective. This representation can inspire more future work. I think the paper should be highlighted at the conference.

---

> > > ### Author Response · Authors · 2025-11-27
> > > **Thank you for the response**
> > >
> > > Thank you very much for your thoughtful engagement and for taking the time to provide further constructive feedback. After reflecting on your suggestions, we will include the clarified explanation for Condition 2, and also adding more qualitative examples with larger camera motion in the revised version. We are happy to elaborate further whenever needed.

---

### Official Review · Reviewer_kaqe · 2025-10-30

**Soundness:** 2
**Presentation:** 3
**Contribution:** 2
**Rating:** 2
**Confidence:** 4

**Summary:**

This paper proposes Trajectory Fields which map each pixel in each frame to a parametric 3D trajectory. To learn the Trajectory Field from videos, the paper proposes Trace Anything module, a feed-forward network to predict control points of B-splines for each pixel in each frame.

To facilitate the training of the proposed model, this paper builds a synthetic video dataset with dense labels, covering 2D/3D trajectories, depth maps, camera poses, and semantic masks.

**Strengths:**

1. It is new to formulate 4D video understanding as pixel-level Trajectory Fields.

2. Collecting a densely labeled dataset for dynamic scene understanding is a very good contribution.

3. The overall presentation is clear and easy to follow.

**Weaknesses:**

1. Modeling 3D trajectories with cubic B-splines over-simplifies the actual movement of 3D points. Complex trajectories cannot be well described with cubic B-splines (p = 3) used in this paper. This means that the proposed method is hard to model complex scenarios.

2. The training of Trace Anything requires dense annotations, which are rarely available in most real-world datasets. When trained solely on synthetic data, the Trace Anything model inevitably suffers from a significant domain gap in real-world applications. This gap encompasses not only differences in geometry and appearance, but also in motion distribution, thereby limiting the scalability of the proposed method.

3. The training objective comprises six terms, yet the paper lacks ablation studies to assess the effectiveness and necessity of each component. Furthermore, these loss functions require extensive additional data annotations, including labels for static and rigid pixels.

4. Experiment settings and results need to be more comprehensive.

4.1. For clarity and reproducibility, it is recommended to explicitly list the training datasets and labels used for all baseline methods.

4.2. In the quantitative comparison on the newly collected synthetic dataset (Trace Anything benchmark), 3D tracking evaluation metrics should be included in addition to trajectory field estimation.

4.3. Quantitative results on real-world datasets should be provided, as this is a crucial evaluation of Trace Anything’s ability to generalize beyond synthetic data.

**Questions:**

1. Given a video of a dynamic scene, is the information sufficient to predict 3D trajectories for every pixel, considering that there are occlusions? If not sufficient, then the motivation of the proposed method is problematic.

2. Since the Trace Anything benchmark evaluates trajectories for pixels sampled from all frames instead of only the first frame, how are the current point tracking approaches adapted to this evaluation protocol?

3. For quantitative comparisons in Table 1, 2, A and B, have other baselines been trained on Trace Anything benchmark dataset? and how?

4. What is the fundamental difference between “all-to-all” evaluation and “first-to-all” evaluation?

---

> ### Author Response · Authors · 2025-11-25
> **Response to Reviewer kaqe**
>
> We thank the Reviewer for the detailed feedback and constructive comments. We appreciate the insights and have updated the manuscript accordingly. Below, we address each point in turn.
>
> **B-Splines Capacity (W1).**  We note that the chosen curve is sufficiently expressive for the general cases we target. As shown in **Table G** and discussed in **Appendix E.8**, a cubic B-spline with $D = 10$ control points models long and complex 3D trajectories effectively and even outperforms a strong 3D tracking baseline. This suggests that the control-point count is not a practical bottleneck in our setting. That said, as noted in our *Limitations*, extremely long trajectories may exceed the capacity of a B-spline, and extending the approach to handle such extreme cases remains future work.
>
> **Real-World Generalization (W2).** In addition to the real-world results shown in the original submission, we include further qualitative and quantitative evaluations on real-world benchmarks and videos in the revised **Appendix E.1 and E.2**, demonstrating the method’s generalizability to real-world scenes. As stated in our *Limitations*, the method is not perfect in generalizing to all real-world settings, and reducing the synthetic-to-real gap remains an important direction for future work.
>
> **Loss Ablation Study (W3).** Please refer to the revised Section 5.3 for the full loss ablation. As shown in **Table 3**, the complete loss configuration provides the strongest results, demonstrating that all loss terms contribute and are jointly important for stable, accurate trajectory estimation. The required annotations come directly from Kubric or from our proposed dataset, so no additional manual labeling is needed.
>
> **Training Data for Baselines (W4.1).** In the revised manuscript, we list the training datasets and labels used for all baseline methods in **Table A**.
>
> **3D Tracking Evaluation (W4.2).** We include 3D tracking evaluation metrics in Table 1 of the revised manuscript.
>
> **Real-World Quantitative Results (W4.3).** Please refer to **Table E** for quantitative results on TAP-Vid3D, **Table B** for KITTI scene flow, and **Table C** for Bonn and TUM-Dynamics. Compared with baselines, our approach achieves best or on-par performance, demonstrating strong generalizability to real-world scenarios across different tasks.
>
> **Information Sufficiency for Prediction (Q1).** In real dynamic scenes, truly “sufficient’’ observation is seldom available; occlusions, partial visibility, and missing evidence are the norm. This defines the challenge and highlights why inferring trajectories from limited observations is valuable in the first place. As with many established CV tasks that operate under incomplete information (e.g., monocular depth estimation, long-term tracking under occlusion, and motion forecasting), full sufficiency of observation is not a prerequisite for developing principled methods. Our model produces reasonable and temporally coherent 3D trajectories even under inherently insufficient observations, for example in **Figure 4** and **Figure N** despite self-occlusions in breakdancing humans or walking elephants.
>
> **Adapting Point Trackers (Q2).**  We query them on all first-frame pixels and use optical-flow matching across frames to associate pixels appearing later in the sequence. If a pixel in a later frame cannot be matched back to the first, we track it forward and backward from that frame separately.
>
> **Baselines trained on our dataset (Q3).** In the original Table 1 and Table 2, the baselines were not trained on the Trace Anything dataset. In the revision, we include **fine-tuned baseline variants** trained with the same scheme as our method on Kubric and Trace Anything datasets; these variants are marked with *. Fine-tuning does improve their performance, but they still fall short of our method. This gap highlights the effectiveness of our trajectory field: its smoothness and temporal coherence yield more stable and accurate motion predictions than point-map–based baselines, even when those baselines are fine-tuned under identical conditions.
>
> **All-to-all vs. First-to-all (Q4).**  All-to-all evaluates trajectories for points sampled from every frame, *naturally covering objects that appear later or undergo visibility changes.* First-to-all only evaluates points visible in the first frame. Thus, all-to-all offers a more complete and reliable measure of multi-frame trajectory consistency.

---

> ### Author Response · Authors · 2025-11-28
>
> Dear Reviewer kaqe,
>
> Thank you again for the thoughtful and constructive feedback. The manuscript has been revised accordingly, and a detailed rebuttal has been submitted in response to all comments. Any further comments or discussion would be greatly appreciated, as they are highly valuable for improving the clarity and quality of the work.
>
> Thank you for the time and consideration.
>
> Sincerely,
> The Authors

---

### Official Review · Reviewer_YYx7 · 2025-10-31

**Soundness:** 3
**Presentation:** 2
**Contribution:** 3
**Rating:** 6
**Confidence:** 4

**Summary:**

This paper proposes a feed-forward network to estimate the 3D trajectories for every pixels in every timestamp, called trajectory fields. To be specific, they use b-splines to model the 3d trajectories, and the control points of the b-spline for each pixel are predicted by a fusion transformer. Thus the each pixel is traced along time. Furthermore, they develop a synthetic data platform, which could benefit training and benchmarking. Several promising results are shown in both video and paired images datasets.

**Strengths:**

1. The proposed model achieves to reconstruct and represent the dynamic monocular videos in a feed-forward manner, making the whole process more efficient.
2. The idea is complete, and shows promising results on two datasets.
3. The model can work on both sequential videos and unsequential paired images.
4. The model proposes an novel framework to fuse the information across different timestamps in one feed-forward inference step.

**Weaknesses:**

1. The presentation is confusing. 1) The authors try to elaborate trajectory field in a formal ‘field’ definition. However, this definition is kind of counter-intuitive. The trajectory field is defined as a function over a discrete sequential image space, but not a trajectory function over space and time, making it hard to understand for many readers, since the mathematical definition always bothers. This would lead to misunderstandings, like someone may assume the trajectory is continuously distributed in the space, but it’s not. Therefore, I kindly request the authors to further elaborate the definition in the next version for easier understanding. 2) The relationship between control points number D and time t is not well discussed. If I understand right, the control points are located uniformly on the spatial temporal line, with k=0 the first frame (t=0) and k=D-1 the last frame (t=1). But the presentation about this part is extremely unclear.
2. Following weakness 1, the definition over discrete space limits the spatial interpolation. In other words, it does not support tracing a new point in the space by the learned trajectories.
3. The experiment is not complete. First of all, since the training requires masks to segment static parts, which is not used for other end-to-end models. So it’s unclear how important this design contributes to the performance. Ablation study is missed here. Second, there is no experiment about motion interpolation along time. For example, given 0, 0.2, 0.4, ..., 1 to predict, eval the model on intermediate time 0.2, 0.3, ... Based on the ablations in Table C, it seems the number of control points do inflence the performance. This shows the temporal interpolation is sensitive to the control point density. Thus this experiment is necessary.
4. The control point number is fixed, which constrains the input video length. Since the motion is smoothly interpolated by b-spline basises, the motion complexity is constrained by the number of control points.
5. The motion forecasting in section 5.3 is not persuasive. The tangent continuation only works in linear constant straight motions, no rotations and accelerations are supported. As for instruction-based forecasting, that ability comes from the video generation model but not trajectory field, thus this should not be claimed as an emergent capability.

I’m open to modify my rating to a large extent based on the authors’ response and other reviewers’ opinions.

**Questions:**

1. How the temporal index embeddings are injected?
2. How the rigid region in equation 12 is determined?
3. Why the correspondence regularization in equation 13 is not enough to regularize static consistency (C1)?

---

> ### Author Response · Authors · 2025-11-25
> **Response to Reviewer YYx7**
>
> We thank the reviewer for the thoughtful and detailed assessment. We appreciate the constructive feedback and have revised the manuscript accordingly. Below, we address each point in turn.
>
> **Trajectory Field Definition (W1.1).** We use the term “trajectory field’’ to denote a field defined directly on the discrete pixel grid. Although this is not a spatially continuous field in the classical physics sense, our usage follows the spirit of the 2D motion field in *Sec. 48.2 of Foundations of Computer Vision (Torralba, Isola, Freeman, 2024)*, where “fields’’ are likewise defined on discrete pixels. In the revised **Section 3.1**, we emphasize that the domain of the trajectory field is discrete and that no spatial continuity is assumed. On the other hand, the codomain is a continuous function of time.
>
> **$D$ and $t$ (W1.2).** This relationship follows directly from the standard definition of spline curves, as outlined in Appendix B. The reviewer’s interpretation $k = 0 \leftrightarrow t = 0$ and $k = D - 1 \leftrightarrow t = 1$ holds and arises from the classical spline definition. In the revised Appendix B, we make this relationship explicit and include a simple 2D illustration in **Figure A**.
>
> **Spatial Interpolation (W2).** Although the trajectory field is defined per pixel and is not inherently continuous in space, one can obtain a trajectory at any off-grid 2D location by bilinearly interpolating the control-point maps to define a valid parametric curve.
>
> **Ablation on Static Regularization (W3.1).** Please refer to the loss-term ablation in **Table 3**. Removing the static regularizer $\mathcal{L}_{\text{static}}$ leads to a clear performance drop, especially in static regions.
>
> **Temporal Interpolation (W3.2).** We include the requested experiment in Appendix E.8. Using inputs at time steps $t = 0, 0.2, 0.4, 0.6, 0.8, 1.0$, we evaluate trajectories both at these given frames and at the unseen interpolated steps $t = 0.1, 0.3, 0.5, 0.7, 0.9$ over 100 sequences. As shown in **Figure Q**, the EPE at interpolated times is only slightly higher than at the given frames, indicating stable temporal interpolation and consistent performance across continuous time.
>
> **Control Point Number (W4).** In most of our targeted cases, our choice of control-point count is not a limiting factor. As shown in **Table G** and discussed in **Appendix E.8**, a cubic B-spline with $D = 10$ control points is already sufficient to model long and complex real-world 3D trajectories, outperforming even the state-of-the-art 3D tracker. We agree that extremely long or highly complex motions may exceed the expressive range of a fixed-$D$ spline, and in such cases, the method can simply be applied in a sliding-window manner without any architectural modifications.
>
> **Motion Forecasting (W5).**   We appreciate the reviewer’s observation. In the revision, we remove “motion forecasting’’ from the claimed emergent capabilities and state only that the method facilitates downstream applications such as simple motion extrapolation.
>
> **Temporal Index Embeddings (Q1).** When frame indices are available, we map the normalized time step to a sinusoidal embedding and add it to all patch tokens. When the index is unavailable, we use a fixed placeholder embedding. We randomly drop time indices during training, so the model learns to handle missing temporal information.
>
> **Rigid Region Determination (Q2).** Kubric contains only rigid motion, so rigid regions are determined directly from its object masks. In the Trace Anything dataset, rigid annotations are limited to the static background, so this term has less effect.
>
> **Equation 13 for C1 (Q3).** Although dense application of $\mathcal{L}_{\text{corr}}$ could, in principle, enforce static consistency, the loss operates on sparse correspondences, and applying it densely would be prohibitively expensive. We therefore include the static regularizer to enforce C1 efficiently by directly penalizing control-point variation over entire static regions.
>
> **Reference:**
>
> Torralba, A., Isola, P., & Freeman, W. T. *Foundations of Computer Vision*. MIT Press, 2024.

---

> > ### Comment · Reviewer_YYx7 · 2025-11-26
> >
> > Thanks for the authors' efforts in significantly modifying the paper, and most of my concerns are addressed, especially the new experiments for ablations and evaluations on real-world benchmarks. I'm delight to increase my score to **8** based on the response. However, I still hold my opinion for the following weaknesses, which hinders me for higher recommendation.
> >
> > 1. Although the authors show the control point number 10 is enough for the selecting benchmark in the appendix, the motion complexity is still constrained. Most of the motion in the demo is continuous and smooth, thus a small number of control point is enough to approximate. But if the motion changes the direction for many times, or shows a fluctuation pattern, the ability of the scheme would be limited. Although the authors say one can use a sliding window to model such complex motions, there lacks a scheme to distinguish where to crop the sequence.
> > 2. The cubic interpolation is struggling in representing accelerations between two consecutive control points. Thus the learned trajectory is not that so physically meaningful. In other words, the time interpolation performance is bounded in acceleration cases.
> >
> > Overall, this is a complete and nice paper, which shows a potential pathway to foundation model in dynamic reconstruction. So I am willing to give a recommendation of acceptance to this paper. Due to the weaknesses above, I can't support it to be score 10 given the limited flexibility in control point number once being trained and the limited capability in handling complex motions.

---

> > > ### Author Response · Authors · 2025-11-26
> > > **Thank you for the response**
> > >
> > > Thank you for the constructive engagement throughout the discussion. We sincerely appreciate your thoughtful guidance, which helped us refine the work and clarify several important aspects.
> > >
> > > Wavy or zigzag-like motion remains a known challenge for dynamic reconstruction and 3D tracking in general. The most straightforward extension for handling such highly oscillatory motion is a fixed-length sliding window, while more adaptive schemes remain an exciting future direction.
> > >
> > > Another promising direction is to further enhance the physical grounding of the trajectory representation. In particular, point-level velocity and acceleration can potentially be controlled and interpreted by linking them to the first and second temporal derivatives of the learned trajectory functions. This could improve robustness under complex motion patterns and enable more physically grounded downstream applications.

---

### Official Review · Reviewer_TmW6 · 2025-11-01

**Soundness:** 4
**Presentation:** 2
**Contribution:** 4
**Rating:** 4
**Confidence:** 4

**Summary:**

The paper introduces a new representation called, Trajectory Fields, for 4D geometry and temporal understanding. Given a 2D video, it reconstructs the dynamic scene in 3D using a set of trajectories. Each trajectory is considered as a small point (or atomic primitive) is moving in 3D space. Trained on a proposed dataset, the method outperforms all existing methods on benchmark datasets

---

The introduced representation is novel and working. However, there are some remaining concerns on missing comparison and unclear details. My initial rating is **4: marginally below the acceptance threshold**, but would like to raise the rating based on the response with a much better understanding of the paper.

**Strengths:**

* **Good, novel representation**

  Existing dynamic 3D reconstruction approaches have a disentangled representation for geometry and motion. To overcome it, the paper proposes a single unified representation, called Trajectory field. In a nutshell, this is similar to dynamic 3DGS but without 3D Gaussian attributes. However, new component proposed in the paper is a controlled point concept that represent the trajectory with a set of controlled points and basis function (eg., B-spline)

* **Good accuracy**

  The proposed method outperforms existing methods by a large margin with the fastest runtime in the proposed benchmarks.

* **Dataset**

  If the proposed dataset is released, it will be a good contribution to the community.

**Weaknesses:**

* **Accuracy on other public benchmark datasets** is missing

  The paper evaluates accuracy on the proposed benchmark only (Table 1 and Table 2), which can be close to the domain that the model is trained on. At least, Table A and B in the supplementary provides accuracy on PointOdyssey and TAP-Vid3D dataset. On PointOdysey, the method performs very competitive as it's on the synthetic domain whereas it has mixed results on the real-world based dataset in Table B. It would be curious to see how the method works on other real-world benchmark datasets (eg., KITTI SceneFlow 2025).

* **Evaluation on 3D pointmap** is missing

  As a related (or downstream) task, the method can naturally output 3D pointmaps. It would be curious to see how the method performs on this 3D pointmap prediction task, comparing to the state-of-the-art reconstruction method, eg., MASt3R, VGGT, and its variant family, also on standard **real-world** dataset such as Bonn, TUM-Dynamics, etc.

* **Confusing notations**

  To be honest, it may need more details to fully understand the equations related to the representation. Here are some questions on the notation for clarification. Assuming we have N input frames (indexed as 0, 1, 2, ..., N-1),

  * Time interval for D? D is the number of control points. Does it mean that there are D control points between each frame pair (eg., 0th and 1st), resulting in N * D total control points in the output sequence? or is it like N frames share the D control points?
  * Instead of using the basis function, what if we assume a linear motion? Does it break the formulation? (not so sure if I understood the concept correctly). Does the accuracy drop significantly?
  * Why is $t$ in Eq. (1) defined [0, 1] not [0, N-1]? Does $t=1$ mean the timestep that corresponds to the last frame (ie, N)? or does it mean any other thing?

* **Missing loss ablation study**

  The paper proposes multiple loss functions from Eq. (10) to (13), but ablation study is missing. It would be good if the paper can provide ablation study and validate the design choice on the loss functions.

* **Overlaying all trajectories from all frames?**

  If the method aggregates all trajectories from all frames $I_i$, this will be quite computationally expensive to represent the scene and can be redundant (eg., it is not like that a very minimal set of atomic primitives move over time and represent the dense 4D scene). How is the method actually implemented? Are there any merging mechanism? (or is this question from my misunderstanding of the paper?)

**Questions:**

* $L_{corr}$ in Eq. (13) seems important for ensuring the 3D consistency of trajectory. What's the loss value look like when the model is converged?

---

> ### Author Response · Authors · 2025-11-25
> **Response to Reviewer TmW6**
>
> We thank the reviewer for the constructive feedback. We value your suggestions and have updated the manuscript accordingly. We address the specific questions below.
>
> **Comparison on Public Benchmarks (W1 & W2).** We compare against VGGT, MonST3R, and MASt3R on real-world benchmarks; see **Appendix E.2** for details.
> - *KITTI Scene Flow 2015:* Our method shows strong short- and long-range performance, accurately capturing object motion (**Figure C**) and outperforming prior work across all metrics (**Table B**).
> - *3D Point Map Estimation:* Our approach preserves geometry and motion boundaries more faithfully (**Figures D and E**), achieving state-of-the-art results on Bonn and competitive performance on TUM-Dynamics (**Table C**).
>
> **Notation Clarification (W3).** Clarifications have been added to **Section 3.1** to prevent ambiguity. Below, we address the confusion:
> 1. Each of the $N$ input frames contributes its own stack of $D$ control-point maps, yielding $N \times D$ maps in total. These maps are not tied to frame pairs and are not shared across frames. Every pixel in frame $i$ receives its own sequence of $D$ control points, which jointly describe its trajectory over the entire normalized time span covered by all $N$ frames.
> 2. Assuming purely linear motion does not invalidate our formulation: using piecewise-linear “hat’’ bases simply connects control points with straight segments, with $D = 2$ reducing to a single global linear segment. However, real object motion is rarely linear, so this choice inevitably reduces accuracy, as confirmed by the experiment in **Table G**.
> 3. We use a normalized temporal domain $t \in [0,1]$ so the formulation is independent of the number of frames. For a video with $N$ frames, $t = 0$ corresponds to the first frame (index $0$) and $t = 1$ to the last frame (index $N-1$). We illustrate $N$ and $t$ in **Figure A**.
>
> **Loss Ablation Study (W4).** Please see the revised Section 5.3 for the loss ablation study. As shown in **Table 3**, the complete loss configuration achieves the best performance, indicating that all loss terms contribute and are jointly necessary for stable and accurate trajectory estimation.
>
> **Overlaying All Trajectories (W5).** We do not merge per-pixel trajectories across frames in our implementation. Saved in the form of per-frame control-point maps, the full representation is only $D$ times the size of the per-frame point maps output by VGGT and related methods, while **users retain the flexibility to select, downsample, or merge them as needed.** In terms of computational efficiency, inference requires only a single feedforward pass. For typical video inputs, Table 1 reports a runtime that is less than one-tenth of the baseline. We further report a typical peak GPU memory usage of approximately 10 GB, also comparable to or lower than existing methods. The reviewer’s comment is helpful: a principled aggregation or merging strategy could further improve accuracy and efficiency for certain tasks, and we view this as a promising direction for future work.
>
> **Correspondence Regularization (Q1).** The loss term $\mathcal{L}_{\text{corr}}$ encourages pixels corresponding across frames to map to the same 3D trajectory, enforcing cross-frame geometric consistency. Table 3 shows that removing this loss degrades performance, confirming its importance. In practice, it stabilizes after roughly $5/6$ of training, converging to about $1 \times 10^{-4}$.

---

> ### Author Response · Authors · 2025-11-28
>
> Dear Reviewer TmW6,
>
> Thank you again for the thoughtful and constructive feedback. The manuscript has been revised accordingly, and a detailed rebuttal has been submitted in response to all comments. Any further comments or discussion would be greatly appreciated, as they are highly valuable for improving the clarity and quality of the work.
>
> Thank you for the time and consideration.
>
> Sincerely,
> The Authors

---

### Author Response · Authors · 2025-11-25
**General Response to All Reviewers**

We thank the reviewers for their time and insightful feedback on our 4D representation paradigm. We are encouraged that the reviewers found our "Trajectory Field" to be a novel and unified representation (**TmW6**) that offers a reasonable formulation for 4D video understanding (**kaqe**, **o94T**), notably without the need for heavy priors (**o94T**).
The reviewers highlighted the efficiency of our approach (**YYx7**), noting its fast inference speeds (**TmW6**, **o94T**) and ability to outperform existing methods (**TmW6**). Furthermore, the community value of our proposed benchmark dataset was strongly recognized (**TmW6**, **kaqe**, **o94T**), and we appreciate that the submission was considered well-written and easy to follow (**kaqe**), with high-quality figures (**o94T**). *We will release the code, pretrained models, and the data platform to support reproducibility and further research.*

We have carefully considered the reviewers' concerns and suggestions. To address them, we have revised the manuscript. A summary of the major updates includes:
* **Clarifications & Scope:** We improved the formulation of Trajectory Fields (Section 3.1) and parametric curves (Appendix B), adding a new illustration in Figure A. We also included a summary of training datasets and labels (Table A) and refined our claims regarding downstream applications.
* **New Comparisons & Baselines:** We added comparisons on real-world benchmarks (KITTI Scene Flow, TUM Dynamics, and Bonn) in Appendix E.1. Additionally, we complemented Tables 1 and 2 with 3D tracking metrics and included baseline results after fine-tuning on the "Trace Anything" dataset.
* **Ablations & Analysis:** We added an ablation study on loss terms (Table 3), an evaluation of temporal interpolation (Figure Q), and a capacity analysis of the chosen parametric curves (Table G).
* **Qualitative Results:** Additional qualitative results have been included in Appendix E.2 to further demonstrate robustness and generalizability.

Below, we provide our detailed responses to each of the reviewers’ concerns.

---

### Meta-Review · Area_Chair_QTnY · 2026-01-08

**Summary:**

Reviewers broadly agreed the paper is novel and efficient, but raised concerns about (i) clarity of the trajectory-field formulation / notation, and (ii) whether results were sufficiently grounded beyond the new synthetic benchmark (missing loss ablations and broader real-world comparisons). The rebuttal/revision directly addressed these by adding clarifications plus new ablations and real-world benchmark evaluations, which moved at least one reviewer to a clear accept. Remaining concerns are largely about inherent limitations (trajectory expressivity for highly oscillatory motion; synthetic-to-real gap / dense supervision assumptions), rather than correctness of the core idea.

**Reviewer Concerns:**

Addressed in rebuttal
(1) Missing ablations / missing benchmark coverage / reproducibility details: authors added loss ablations, real-world benchmark comparisons, and clearer dataset/label listing.
(2) Notation / definition confusion (trajectory field domain/codomain, control points, etc.): authors state they revised Section 3.1 / Appendix and added an illustration.
(3) Over-claiming motion forecasting: authors explicitly toned down the claim to “simple motion extrapolation.”

Still outstanding
(1) Modeling highly oscillatory / very complex motion with fixed cubic B-splines: authors acknowledge sliding windows / future directions, but this remains a limitation.
(2) Synthetic-to-real generalization and dense-label reliance: authors added more real-world evaluation, but the reviewer’s scalability/domain-gap concern is structural and not fully eliminated.
(3) No principled trajectory “merging/aggregation” mechanism (efficiency/redundancy): authors note this as future work.

**Reviewer Scores:**

o94T: likely unchanged at 10 (strong accept; wants even more quant for large camera motion but still “should be highlighted”).

YYx7: increased to 8 explicitly after rebuttal and new experiments.

TmW6: from 4 (“marginally below threshold”) to ~6 is plausible given the revision added the exact items requested (public benchmark comparisons, loss ablation, clarification of implementation).

kaqe: likely remains below the bar but could move from 2 toward ~4 given added real-world eval + loss ablation; core concerns (expressivity and synthetic-to-real / dense labels) still stand.

---

### Decision · Program_Chairs · 2026-01-26

Accept (Poster)